

# New approaches to dating intermittently varved sediment, Columbine lake, Colorado, USA

Stephanie H. Arcusa[1], Nicholas P. McKay[1], Charlotte Wiman[2], Sela Patterson[1], Samuel E. Munoz[2,3], Marco A. Aquino-López[4]

[1]School of Earth and Sustainability, Northern Arizona University, Flagstaff, 86011, USA
[2]Department of Marine and Environmental Sciences, Northeastern University, Marine Science Center, Nahant, 01908, USA
[3]Department of Civil and Environmental Engineering, Northeastern University, Boston, 02115, USA
[4]Centro de Investigación en Matemáticas (CIMAT), Jalisco s/n, Valenciana, 36023 Guanajuato, Gto, Mexico

*Correspondence to*: Stephanie H. Arcusa (sha59@nau.edu)

**Abstract.** Annually laminated lake sediment can track paleoenvironmental change at high-resolution where alternative archives are often not available. However, information about both paleoenvironmental change and chronology are often affected by indistinct and intermittent varves. We present an approach that overcomes these and other obstacles by using a quantitative varve quality index combined with a multi-core, multi-observer Bayesian varve sedimentation model that quantifies realistic under- and over-counting uncertainties while integrating information from radiometric measurements ($^{210}$Pb, $^{137}$Cs, and $^{14}$C) into the chronology. We demonstrate this approach on thin sections of indistinct and intermittently varved sequences from alpine Columbine Lake, Colorado. The integrated model indicates 3137 (95 percentile highest density probability range: 2753-3375) varve years with a cumulative posterior distribution of counting uncertainties of -13/+7 % indicative of systematic observer undercounting. The sedimentary features of the thin and complex varves shift through time, from normally graded couplets to couplets interrupted with coarser sub-laminae, to inversely graded couplets. We interpret the normal grading couplets as spring nival discharge followed by winter settling, the coarser sub-laminae as high rainfall events, and the inverse grading as hyperpycnal flows and/or pulses of dust related to human impact changing the varve formation mechanism. Our novel approach provides a realistic constraint on sedimentation rates and quantifies uncertainty in varve counts by quantifying over- and under-counting uncertainties related to observer bias and the quality and variability of the sediment appearance. The approach permits the construction of a varve chronology and sedimentation rates for sites with intermittent or indistinct varves, which are likely more prevalent than sequences with distinct varves, and thus, expands the possibilities of reconstructing past environmental change with high resolution.

## 1 Introduction

The establishment of a reliable chronology for lake sediment is a pre-requisite of paleoenvironmental investigation. As many studies have pointed out, low age uncertainty is necessary to compare events through space, time, and archive (Zimmerman and Wahl, 2020). To that end, annually laminated sediment (i.e., varves) not only presents a unique opportunity to





reconstruct variability on a seasonal to annual scale, it allows for the quantification of sediment accumulation rates on shorter timescales than sequences dated by radiometric techniques (Boers et al., 2017). Sedimentation rates are useful for a wide range of investigations, but especially so for the accurate calculation of fluxes (g cm$^2$ yr$^{-1}$) of sedimentary constituents. For paleoenvironmental reconstructions, flux is typically a more meaningful measure than abundance or concentration

because it considers changes in the sediment due to time and density. For example, investigations using lake sediment of past aerosol deposition such as dust report different conclusions when flux is used compared to abundance (Arcusa et al., 2019; Routson et al., 2016, 2019). The importance of constraining age and sedimentation rate uncertainty is increasingly recognized and the tools to handle this uncertainty are constantly improving (Aquino-López et al., 2018; McKay et al., 2020).

Despite general improvements, the quantification of uncertainty in varved sediments remains focused on counting. Although there is no standard method for calculating uncertainties in varve chronologies, most are associated with ±1-4 % counting uncertainty with some indistinctly varved sequences having counting errors up to ±15 % (Ojala et al., 2012). Counting errors are often quantified as the root mean squared error of counts from multiple observers along defined transects on multiple

cross-dated cores from the same site either as maximum and minimum deviations from the mean or as replicated counts between marker layers (Lamoureux, 2001). Reported error estimates commonly do not include all known error sources.

Error sources are associated with (1) inter-site differences in varve counts (missing varves), (2) subjectivity in identifying varves due to varve quality, (3) expert judgement in identifying marker layers, (4) compound single varves that are mis-

interpreted as representing multiple years (over counting), (5) indistinct varves that are combined with adjacent varves (under counting), (6) intermittent (floating) varves, (7) technical issues (missing varves), and (8) counting strategies (Fortin et al., 2019; Ojala et al., 2012; Żarczyński et al., 2018; Zolitschka et al., 2015). Although these various sources are often considered individually, they are less frequently considered in concert and rarely considered when estimating sedimentation rates. The variety of error sources makes their quantification an important challenge, especially for sequences with indistinct

or intermittent varves.

Sedimentary sequences with indistinct or intermittent varves cannot be used to develop a chronology with conventional techniques as the massive sediment or indistinct laminations result in information loss. The problem is often addressed by subjectively applying the sedimentation rate estimated from neighboring varved sections, although more mechanistic

methods have also been developed. For example, Schlolaut et al. (2012) describe a procedure that analyses the seasonal layer distributions to estimate the number of years of sediment accumulation represented. Although promising, such a method of varve interpolation has yet to be integrated with a complete accounting of all other errors.





Few previous works have attempted to assess errors associated with varve counts by their sources. For example, Fortin et al.
(2019) developed a Bayesian probabilistic model to incorporate three sources of uncertainty related to the subjectivity in
identifying varves, inter-site differences, and a combination of the likelihood of over- and under-counting by the observer
and the proper identification of isochronous marker layers. Although this model provided a clearer picture of the sources of
uncertainty, it did not go as far as addressing the problem of indistinct varves (such as those deposited during the 20th
century as glacier influence waned) nor quantifying the impact of varve quality on the chronology.

Additionally, errors can be systematic in that the net outcome is one of over- or under-counting. These systematic biases are
typically assessed by comparing the varve chronology to radiometric methods ($^{137}$Cs, $^{210}$Pb, and $^{14}$C) and can sometimes be
corrected. For example, the agreement between varve and radiometric chronologies can be evaluated objectively through
OxCal's V_sequence, for example (Bronk Ramsey, 1995; Tian et al., 2005; Zander et al., 2019). The $^{14}$C ages can reveal
missing sediment intervals where missing varves can be inserted (Tian et al., 2005). However, the process has two major
drawbacks. First, the $^{14}$C ages could be too old, or, if they are correct, the location of the nonconformity in the sedimentary
sequence might be misplaced. Second, this approach does not constrain the uncertainty introduced into the estimation of the
sedimentation rate.

Here, we present an approach to quantify age and sedimentation rate uncertainty using multiple cores and observers as
demonstrated in a case study of an indistinctly and intermittently varved sequence from Columbine Lake, Colorado. We
expand on the Fortin et al. (2019) Bayesian model to include uncertainty from multiple observers, varve interpolation, and
varve quality. We then use Bayesian learning to update prior estimates of the counting uncertainties given the constraints
from independent radiometric ages. Partly because continuous chronologies are rare, no late Holocene varve sequence has
been published from the southern Rocky Mountains up to now. Moreover, the nearest published varved lake record is 250
km away (Anderson et al., 2010). The chronology developed here provides the foundation for future high-resolution
paleoenvironmental research at Columbine Lake.

## 2 Study Site

Columbine Lake (37.8622° N, 107.7717° W, elevation 3874 m) is a deep, mildly acidic (pH 5), oligotrophic lake in San Juan
County, Colorado (Fig. 1). The lake bathymetry is marked by deep pockets, with a maximum depth of 27 m. Deep and small
sub-basins favor seasonal stratification and anoxic conditions necessary for varve formation (Zolitschka et al., 2015). The
lake is fed by a small pond and stream to the northwest and drained by Mill Creek to the northeast. The catchment bedrock is
andesite emplaced during the late and middle Tertiary (Lipman and Mcintosh, 2011), and less than 5 % of the area was
vegetated in 2017 (Arcusa et al., 2019). The catchment is currently unglaciated and shows no evidence for rock glaciers. The



closest documented evidence of a Little Ice Age moraine is near Trinity Peaks (Carrara, 2011). There are no access roads,
    but historic mining activity is evident at lower elevations and the lake outflow is raised by a 2-m-high earthen dam.

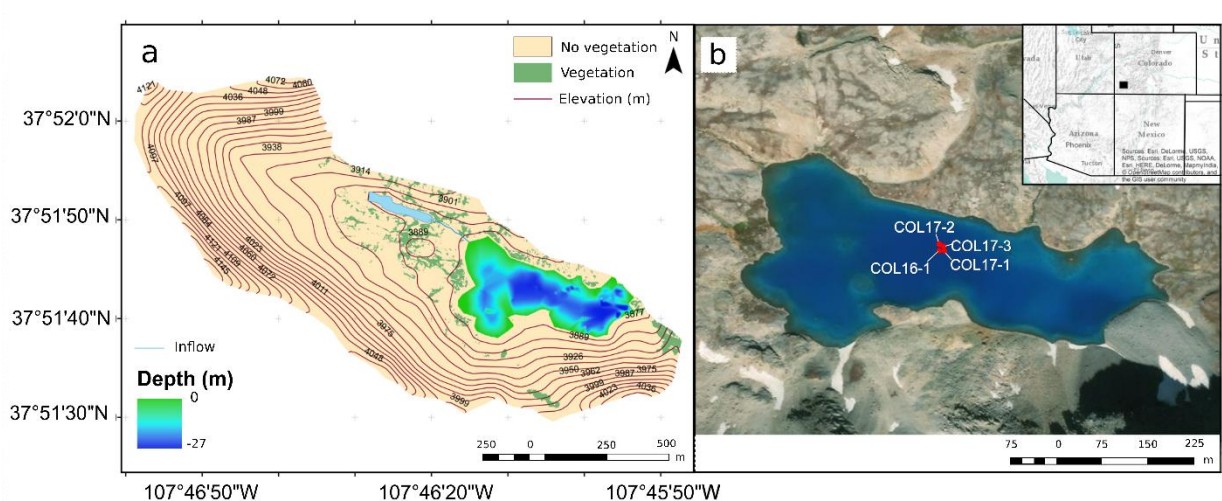

**Figure 1. Columbine and its catchment showing (a) bathymetry and (b) coring location (red circles) in southwest Colorado (black rectangle in inset map). Vegetation extent for the year 2017 based on Arcusa et al. (2019). Image**
**credit: Esri, DigitalGlobe, GeoEye, Earthstar Geographics, CNES/Airbus DS, USDA, USGS, AeroGRID, IGN, and the GIS User Community.**

The climate of Silverton, Colorado (elevation 2865 m) near the study site is typified by a biseasonal climate. Over 80 % of precipitation falls predominantly as snow from October to March (average 560 mm/month total snowfall) from Pacific
frontal storms, and summer rainfall is associated with the northern extent of the North American Monsoon (Jul-Sep, average 70 mm/month). Average winter (DJF) and summer (JJA) temperature ranges are –18.8 to 2.5 °C and –0.1 to 22.8 °C, respectively (Western Regional Climate Center, 2018). Like much of the Southwest United States, the El Niño Southern Oscillation (ENSO) teleconnection usually results in wet winters during El Niño and dry winters during La Niña (Sheppard et al., 2002).

**3 Methods**

**3.1 Coring**

Four sediment cores were collected from Columbine Lake at water depths ranging from 25 to 27 m. One 81-cm-long core was taken in August 2016 (COL16-1) using an aquatic corer, and three 125- to 142-cm-long cores were collected in September 2017 (COL17-1, COL17-2, and COL17-3) using a modified UWITECH percussion coring system. All three 2017
cores captured the undisturbed sediment-water interface, but the 2016 core did not. Cores were split, described, and stored at



the Sedimentary Records of Environmental Change Lab at Northern Arizona University. Consistent core stratigraphy and marker layers found in all cores except COL17-1 facilitated visual core cross-correlation (**Error! Reference source not found.** Fig. A1). Core COL17-1 is not laminated, possibly because it was collected at shallower depth, and was not considered further in this study.

**3.2 Non-destructive core analysis**

To support the visual correlations, cores COL16-1, COL17-2, and COL17-3 were analyzed for non-destructive methods. First, magnetic susceptibility (MS) was measured at 1 cm increment (2 cm measurement diameter resolution) using a Bartington MS2 surface sensor. Then, X-Ray Fluorescence (XRF) was measured at 0.5 and 1 cm intervals (1 cm measurement diameter resolution) at 10, 30, and 50 kV using an Avaatech core scanner at Texas A&M University, College Station. Finally, hyperspectral imaging in the visible to near-infrared range was measured at ~68 µm/pixel using a Specim Ltd. core scanner equipped with a PFE-xx-V10E camera at Northern Arizona University following the method by Butz et al. (2016). The hyperspectral data were used to calculate indices shown to be related to chlorophyll and its degradation products (RABD660) (Trachsel et al., 2010; Yackulic, 2017) as well as chlorite (minimum peak) (Rein and Sirocko, 2002).

**3.3 Destructive core analysis**

To support the sedimentological facies interpretation, various destructive analytical analyses were performed on core COL17-3. Loss-on-ignition and wet and dry bulk density following Dean (1974) used 1-2 cm$^3$ of sediment weighed wet and dry after freeze-drying for 12 hours, then weighed after burning at 550 ºC for 5 hours in the furnace. An aliquot of 80 mg of material was then used for quantifying the abundance of biogenic silica following an adapted procedure of Mortlock and Froelich (1989). Briefly, the samples were pre-treated to remove organics. Biogenic silica was brought to a solution and measured by spectrophotometry. Finally, an aliquot of 200 mg of material was used for grain size analysis. The initial procedure was the same, but the solution of biogenic silica was discarded. Then, sodium hexametaphosphate was added as a dispersant and shaken for 3 hours. Grain size distributions in the 0.04–2000 µm range with 116 classes were analyzed using a laser diffraction Coulter LS13-320 and each sample was measured 5 times.

**3.4 Geochronology**

This study added three radiocarbon dates to the three previously published by Arcusa et al. (2019) on cores COL17-3 and COL16-1. Macrofossil of terrestrial plants and aquatic insects were pre-treated using standard acid–base–acid procedures and analyzed for radiocarbon activity on Northern Arizona University's MICADAS equipped with the Gas Interface System while it was located at the manufacture's (IonPlus) office in Zurich, Switzerland. In addition to radiocarbon, Arcusa et al. (2019) also measured $^{210}$Pb and $^{137}$Cs activities respectively on 20 and 16 dried and homogenized samples over the top 12.5 cm of core COL17-3 using a Canberra Broad Energy Germanium Detector (BEGe; model no. BE3830 P-DET) at the Marine Science Center at Northeastern University.



The radiometric age-depth model was constructed from the concurrent use of Bayesian modeling R software packages Bacon (Blaauw and Christen, 2011) and Plum (Aquino-López et al., 2018). Briefly, Plum is based on a statistical framework, which uses statistical inference to provide more robust and realistic uncertainties when compared to the Constant Rate of Supply (CRS) method (Appleby and Oldfield, 1978). The concurrent use of Bacon and Plum reduces the artificial break in sedimentation rates at the intersection of the $^{210}$Pb and $^{14}$C ages, and Plum provides a more natural merger of these techniques as it does not require the pre-modeling of the $^{210}$Pb dates. Additionally, we compare Plum to conventional calculations of CRS (Appleby, 2001) and the Constant Flux Constant Sedimentation (CFCS) method (Krishnaswamy et al., 1971) implemented with the R package SERAC (Bruel and Sabatier, 2020).

### 3.5 Thin sections, sediment imaging, and point measurements

To facilitate investigation, measurement, and counting of the fine laminations, the sediment was subsampled and impregnated with low viscosity epoxy resin following a modified approach of Lamoureux (1994). The percentage of epoxy to acetone was increased multiple times before fully embedding the sediment. Overlapping sediment slabs (7.0 x 3.0 x 1.5 cm) were sampled and placed in an acetone bath for fluid replacement. Acetone was exchanged every 12 hours for five days until no water was left in the sediment. Following fluid displacement, Spurr's Low Viscosity Embedding Resin was exchanged every 12 hours for three days and left to cure for one day at room temperature followed by one day at 40 ºC, one day at 50 ºC, and one day at 60 ºC. Slabs were cut at the Northern Arizona University machine shop and sections were sent to Quality Thin Sections in Tucson, AZ, for mounting and polishing. Images of the thin sections were taken at 2x and 10x magnification under polarized light with a calibrated petrographic polarizing microscope (Carl Zeiss Axiophot) connected to a digital camera (Carl Zeiss Axiocam) and automated stepping stage (PETROG System, Conwy Valley Systems Ltd (CVS), UK). Individual images were stitched into a mosaic using the Stitching plugin (Preibisch et al., 2009) in ImageJ.

To categorize and interpret varve facies, microscopic analyses of elemental composition and grainsize are sometimes used (Cuven et al., 2010; Żarczyński et al., 2019a). In this study, the varves were thinner than the sampling resolution of either destructive (BSi and grain size) or non-destructive (XRF, hyperspectral, and MS) procedures available. Therefore, we used point counts and length measurements directly on individual grains in the slides. At least 100 grains were measured from the varve transects.

### 3.6 Statistical analyses

To support the interpretation of the sedimentary facies, statistical analyses were performed on the results from both destructive and non-destructive procedures. First, the values were binned to match the sampling resolution of the dataset with the lowest resolution using the function bin2d in the R package geoChronR (McKay et al., 2021). Second, the values were standardized to a mean of zero and variance of one standard deviation. Then to identify distinct stratigraphic units,



hierarchical cluster analysis was applied using the function chclust R package rioja (Juggins, 2020). To associate units to the
variables explaining the most variance, a principal component analysis that was applied with the function PCA in the R
package FactoMineR (Lê et al., 2008). Finally, to explore the relationship between variables, correlation analysis was
performed using Spearman's rank as the data distribution failed the Shapiro-Wilks normality test in most cases ($p < 0.05$).

### 3.7 Varve chronology

### 3.7.1 Description of the original varve model

The data analysis in this study expands on the original R (R Core Team, 2019) package varveR (McKay, 2019) that builds
varve chronologies while quantifying uncertainty as it relates to varve identification, inter-site differences, and likelihoods of
over- and under-counting. varveR is a Bayesian probabilistic model that quantifies age uncertainty by integrating
information from the age distribution of marker layers from multiple cores (Fortin et al., 2019). The model follows two
concepts. First, it uses the sedimentological understanding of the likelihood of the correct delineation of the varves such as
those related to the ease of distinguishing them. Second, it takes advantage of the replication from the marker layers
correlating between cores to quantify the likelihood of under- and over-counting and the uncertainty in the total count as a
function of depth.

The model's inputs include (1) thicknesses for each varve for each core, (2) site-specific marker layers to stitch the thin
sections together into a varved sequence, and (3) inter-site marker layers. In this study, thickness delineations were created
as ArcMap shapefiles. Site-specific marker layers were identified in the overlap between two adjacent thin sections. Inter-
site marker layers were identified in each core for cross-correlation. All three were identified by three observers working
independently to explore uncertainties associated with expert judgment.

The model uses prior likelihoods of over- and under-counting and updates them as it iterates. The prior likelihoods are
selected by the operator but may be the difference in the number of varves counted by two observers expressed as a
percentage and converted into a probability, for example (Fortin et al., 2019). With each iteration, the only constraint is that
the duration across cores between marker layers must be the same. varveR outputs an n-member ensemble of varve counts
and thicknesses for each core and a composite of all cores, where n is a user-defined number of iterations. The ensemble is
used to quantify the uncertainty in depth as a function of varve year and can be transposed to estimate uncertainty in varve
year as a function of depth. The model is completely independent from radiometric age control.

### 3.7.2 Modifications to the original model: varve quality index and varve emulator

We expanded the varveR model to include information on varve quality as an indicator of the likelihood of over- and under-
counting. Although varve quality indices have been used in past research as a qualitative aide to interpretation (Bonk et al.,



2015; Dräger et al., 2017; Żarczyński et al., 2018), here we integrate this information quantitatively. Each varve was associated with a code (1, 2, or 3) (Appendix A Fig. A2) with a corresponding distribution of over and undercounting prior probability estimate (Sect. 3.7.3). The codes are assigned by the clarity of the varve's appearance, with a code value of 1 being of higher clarity than a code value of 3. A code of 4 was used when it was difficult to distinguish whether two couple represented one or two years. In this case, they were counted as two varves, and denoted with a code of 4, which were

assigned a 50 % probability of over-counting. The application of code 5 is described below. Finally, sections where sediment is likely missing for technical reasons (e.g., between two adjacent thin sections without overlap or in gaps created during the embedding process), were assigned a code of 6, and varves were similarly emulated although the number of missing years is unknown.

Distinctly varved sediments are interspersed with indistinctly varved sections, which comprise zones up to 2 cm thick with weakly defined to no visible laminations (Appendix A Fig. A2). These indistinct sections were relatively common, comprising 8.7-19.6 % of the total sediment thickness across observers. For these sections, a code of 5 was assigned. Previous studies have addressed the issue of indistinct varve sections by either interpolating sedimentation rates from nearby varved segments (e.g. Hughen et al., 2004), or using the probability distribution of the varves' seasonal layers to derive

sedimentation rates (Schlolaut et al., 2012). Because our varveR approach requires an estimate of varve thicknesses for each year rather than an estimate of mean sedimentation rate or missing time, these solutions are insufficient. Instead, we simulate varves through these sections.

To simulate varves in indistinct intervals, we developed a varve emulator that randomly chooses a distinctly varved section

of the core and with a length of that section matches the thickness of the interval as nearly as possible. Because laminations at Columbine Lake are very thin (typically < 0.5 mm) relative to the thickness of the indistinct intervals (typically ~ 4 mm), this procedure alone matches the cumulative depth closely. Subsequently, a minute thickness adjustment is applied across the sequence to ensure a perfect match in total thickness and conservation of the depth of the core. This approach is reasonable where other varved intervals can serve as reasonable surrogates for indistinct sections. We argue this is the case for

Columbine Lake, as the distribution of the varve thickness is similar in both cores throughout the sections with distinct varves (Appendix A Fig. A3). Furthermore, there is no evidence for systematic changes in the mode of deposition in these sections, as the indistinct sections occur throughout both cores, but not always at the same time and the sedimentary features were mostly the same above and below the indistinct sections.

### 3.7.3 Chronology with symmetrical and asymmetrical uncertainty

The modified varveR model was used to build two varve chronologies each following a different scenario. In both scenarios, codes 1, 2, and 3 were given over- and under-counting priors. In the first scenario, the priors were symmetrical and based on values found in the literature (Fig. 2a; e.g. Dräger et al., 2017). This was done to produce a chronology that would resemble




the conventional varve chronology construction and allow for comparison. However, due to missing or indistinct varves, varve chronologies are often subject to under-counting (Tian et al., 2005; Żarczyński et al., 2018). Because the varves in this lake are thin and often lacked clarity in their appearance, we considered a symmetrical prior to be unrealistic for Columbine Lake. A prior shifted towards under-counting was deemed more representative. Therefore, in a second scenario, we assigned wider symmetrical priors for code 1, wider asymmetrical priors for code 2, and assigned an uninformed asymmetrical prior for code 3 (Fig. 2b). This expanded version of varveR incorporates uncertainty pertinent to varve quality, inter-site variation, expert judgment (Fig. 3).

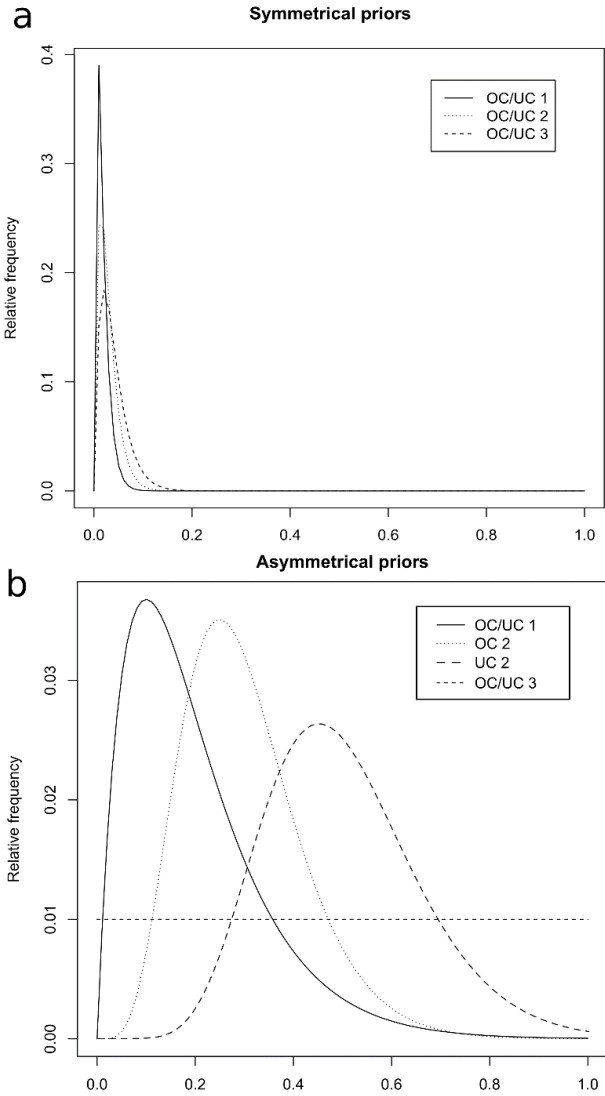





**Figure 2. Varve quality codes and their associated under- (UC) and over-counting (OC) gamma distribution priors for (a) symmetrical and (b) asymmetrical varveR.**

### 3.8 Varve chronology verification

A varve-based age-depth determination should be cross-checked with other independent dating methods to (1) support the interpretation of varves as annual and (2) to identify systematic errors (Ojala et al., 2012; Zolitschka et al., 2015). To do so, the varveR and integrated model output is depth-calibrated and displayed as age-depth curves. Then, the near-surface counts are compared to radionuclide ($^{137}$Cs and $^{210}$Pb) based age-depth models that use conventional CRS and CFCS and Plum, a Bayesian approach to $^{210}$Pb dating (Sec. 3.4). The full sequence is compared to a Bayesian radiocarbon age-depth model. All
comparisons are made using the dated core (COL17-3).

### 3.9 Varve and radiometric chronology integration

Bayesian statistics provide the opportunity to combine different chronological data and their uncertainty (e.g. Buck et al., 2003) as well as information regarding the sedimentation process (e.g. Blockley et al., 2008) by informing priors (Brauer et al., 2014). Here we use Bayesian learning to update prior estimates of the counting uncertainties for each observer given the
constraints from the independent radiometric model. Then, we combine the model into a master chronology.

Our Bayesian framework uses a custom Gibbs sampler to improve on the prior estimates of likelihood probabilities of over- and under-counting described for the varveR model. The Gibbs sampler is initialized using the prior estimates of over- and under-counting used in asymmetrical varveR (Fig. 2b). The sampler updates using an objective function that calculates the
likelihood of a proposed varve chronology given the radiometric ages and their probability distributions. We assume the probabilities associated with varve quality codes 1 and 2 are best described using gamma distributions and must fall between 0 and 1. For algorithmic efficiency, we loosely impose the assumption that proposed adjustments that increase over-counting rates should be balanced by decreases in under-counting rates, although overall reductions in both over- and under-counting are possible and do occur. The output of the log objective function is the product of the age probabilities of all radiometric
samples and the over- and under-counting likelihood of all varve quality codes. The higher the output value, the closer the improved varve count is to the maximum likelihood of the product of the radiometric ages. The Gibbs sampler innovates on the previous over- and under-counting probabilities with each iteration if by adjusting with a small random number from a normal distribution if there is an improvement in the output of the log objective function (i.e. a higher value). We ran the Bayesian algorithm independently for each of the three observers until the objective values stabilized (~100 iterations), then
removed the burn-in and thinned the parameter chain to keep 1000 values. Finally, for each observer, we select the parameters corresponding to the 300 highest objective values and combine them into combined posterior distributions. These posterior distributions on the counting rates are then used to drive an updated varveR model and produce a master chronology that effectively combines the radiometric model and the varve measurements from all observers (Fig. 3).





**Figure 3. Schematic of the approach used in this study. (1) Gathering raw measurements of varve thickness, counts, and marker layers for each core and each observer. (2) Using a modified version of varveR to produce a chronology following scenario 1 (symmetrical and literature-derived likelihoods of over- and under-counting) and scenario 2**

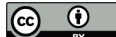



(asymmetrical and larger likelihoods of over- and under-counting). (3) Integrating radiometric information into the varve chronology by updating the prior likelihoods of over- and under-counting in an objective function. The posteriors of the nth best function output are used to run varveR and produce the final chronology that minimizes systematic bias and quantifies uncertainty related to misidentifying marker layers, observer bias, and varve quality and outputs sedimentation rates with uncertainty.

## 4 Results

### 4.1 Sediment profile

Columbine Lake sediments were previously described generally by Arcusa et al. (2019) and more detail is provided here. The sediments contain five stratigraphic units composed of minerogenic, laminated silts and clays ranging in color from grey to reddish-brown to orange (Fig. 4 and Fig. 5). Three of the four cores showed identical sediment profiles, but only COL17-2 and COL17-3 captured an intact sediment-water interface and laminations (Appendix A Fig. A1).

#### 4.1.1 Units 5 and 4

Unit 5 (141-126 cm; depths in core COL17-2) is characterized by massive grey clay-sized sediment and lithogenic indicators (Si, Ti, K, Al, Rb, MS) are typically high and covary (Appendix A Fig. A4, A5 and A6). Unit 5 contained missing data so could not be included in the PCA (Fig. 5). The transition between units 5 and 4 is marked by a large and rapid increase in the redox element Mn, along with an instantaneous increase in Mn/Fe (Appendix A Fig. A5). Unit 4 (123-108 cm) is the first unit to contain laminations and correspond to the most elevated Fe and P. This unit contains type 1 varves.

#### 4.1.2 Unit 3

Unit 3 (105-75 cm) contains poor quality laminations frequently interspersed with indistinct sections. The sections of indistinct varve preservation generally correlate across the parallel cores, although are more prevalent in core COL17-2 (Fig. 4). Unit 3 is characterized by type 1 varves.

#### 4.1.3 Unit 2

Unit 2 (72-12 cm) contains laminations of average clarity with indistinct sections (Fig. 4a). The sections of indistinct varves generally correlate across the parallel cores, with exceptions. Type 1 varves are present, although type 2 varves start to appear intermittently in core COL17-2. The break between units 3 and 2 coincides with a general shift from type 1 to type 2 that is evident both in the thin section microfacies analysis and the hierarchical clustering. Unit 2 sees a small but significant decrease in magnetic susceptibility and Fe compared to unit 3 (Appendix A A4 and A5).





### 4.1.4 Unit 1

This unit (12-0 cm) contains well-defined laminations as well as massive fine silt layers and can be further split into two sub-units. The lower sub-unit (12-2 cm) contains fine, grey, type 3a and b varves interspersed by two massive layers. The two massive light brown layers are both in core COL17-2, with core COL17-3 only containing the youngest of the two. Core COL17-3 contains a layer of indistinct laminations that cross-correlates with the oldest of the two COL17-2 massive layers suggesting the layers are composed of poorly preserved varves as opposed to single massive bed deposited rapidly. The other sub-unit (0-2 cm) contains thicker bright orange type 3b varves just below the sediment-water interface. Organic and biogenic (percent organics, biogenic silica, and green pigments as indicated by the index of RABD660) abundance increase to their highest levels in the top sub-section (Fig. 5, Appendix A Fig. A5), indicating increased lake productivity. Some heavy metals (Zn, Ag) also increase to their maximum levels (Appendix A Fig. A4).



**Figure 4. Sediment, proxy, and varve profiles. (a) Lithostratigraphy and location of radiometric samples of cores COL17-3 and COL17-2. Images are true color. The base of COL17-3 is black because the oxidized red crust has been scraped off. MS = magnetic susceptibility, BSi = biogenic silica. Other proxies are shown in Appendix A A5. (b)**






**Microscopic thin section examples of varve types 1, 2, and 3. (c) Microscopic sub-lamination grain size analysis of**
**varve types 1, 2, and 3b.**

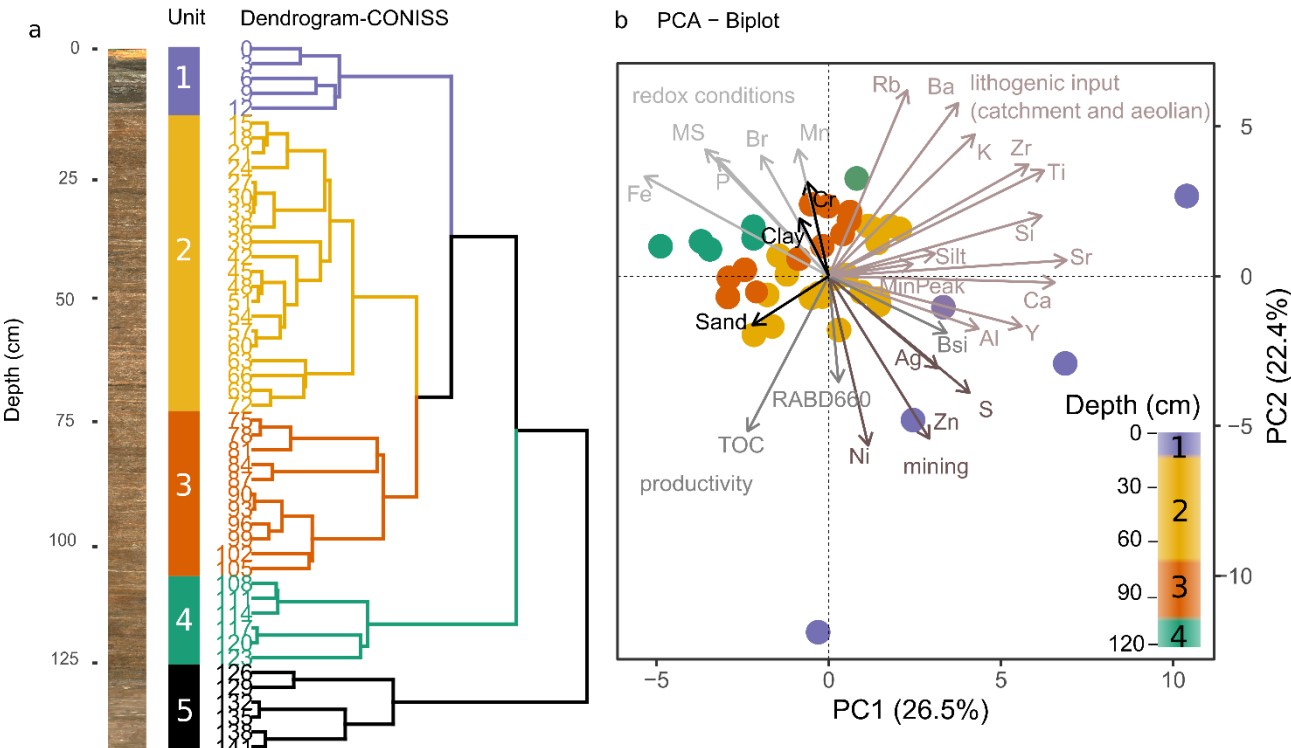

**Figure 5. Statistical analysis of proxy data from core COL17-3. (a) A constrained dendrogram with significant clusters representing the stratigraphic units (1-5) color-coded and applied to the sample depths used in the (b) Principal Component Analysis (PCA) biplot for reference. The first two principal components explain 48.9% of the**
**variability. PC variables grouped by indicator type have different colors. The image of the core is presented for context. PCA loadings and scores can be found in Appendix A Fig. A7.**

### 4.2 Varve type description

The examination of thin sections revealed complex microfacies that repeat within each lamination, indicative of a rhythmic change in the depositional environment. Moreover, comparison to radiometric measurements demonstrate this rhythmic
layering is annual (Sect. 4.6). Therefore, the sediment is described here as true non-glacial clastic varves. Three main types of clastic varves are further sub-divided based on their internal structure (Fig. 4b). Type 1 is composed of typical couplets of silt and clay, type 2 couplets are interrupted by a third coarser grained sub-laminae, and type 3 couplets are inversely graded (Fig. 4c).



### 4.2.1 Type 1

Type 1, most common in the deepest half of the sequence, consists of couplets identified by color and grain size. The bottom part, lithozone I, is characterized by ungraded or fining upward grading of light reddish-brown sediment with grains that measure 5-15 μm (Fig. 4c). The top part, lithozone II, is a fine-grained, dark-brown clay-rich cap with grains consistently < 5 μm (Fig. 4c). The contact between lithozones I and II is generally gradual.

### 4.2.2 Type 2

Varve type 2 is most common in the top half of the sequence and consists of couplets (lithozone I and II) interrupted by coarser-grained (25-40 μm) matrix-supported sub-laminae (lithozone III). An erosional contact separates lithozone I from III, which is composed of plagioclase, quartz, and oxides, as identified under polarized microscope light. Like type 1 varves, type 2 varves are terminated with a dark reddish-brown clay cap (<5 μm, lithozone II).

### 4.2.3 Type 3

Type 3 varves are found exclusively at the topmost part of the sequence and can be sub-divided into varve type 3a and 3b. The deepest of the two, type 3a, is generally thicker and contains lithozone IV. Lithozone IV is characterized by a reverse grading of fine and dark grains at the bottom to coarse and light sediment at the top (Fig. 4c). Lithozone IV is followed by a thin and sometime non-existent lithozone II. Finally, at the topmost part of the cores is varve type 3b, similar in composition to varve type 3a. The difference is a strongly pronounced clay cap (lithozone II). Varve type 3 differs from type 2 because

the coarsest grains appear gradually within lithozone IV rather than abruptly in lithozone III. Lithozone IV in varve type 3a and b also gradually change in color from dark to light.

### 4.3 Varve counts, thicknesses, and quality

Varve thicknesses, excluding varves of quality code 4, 5, and 6, are similar for each core (Table 1), with a combined mean and standard deviation of 0.5 ± 0.05 mm. Thick varves were found in COL17-3. Varve quality was generally higher at the

top of the two cores (code 1) and fluctuated between moderate and poor quality throughout (Fig. 6).

With symmetrical varveR, cores COL17-2 and COL17-3 contain a total of 2466 (highest probability density region: 2075-2880) and 2380 (1999-2710) varves, respectively (Table 2, Fig. 7). This amounts to a cumulative uncertainty of -391/+414 varves (-17/+15 %) for COL17-2 and -381/+330 (-17/+13 %) for COL17-3. With asymmetrical varveR, the mean total varve

count increases by 300-400 varves to 2865 (1417-3923) for COL17-2 and 2740 (1394-3742) for COL17-3 although the cumulative uncertainty also increases to -1448/+1058 varves (-68/+31 %) and -1346/+1002 varves (-65/+31 %), respectively.





**Table 1. Summary statistics for varve thicknesses based on the average of all observers' measurements, excluding intervals of indistinct laminations. Total varve counts indicate output of symmetrical varveR.**

| Core | COL17-2 | COL17-3 |
|---|---|---|
| Length of varved sequence (cm) | 127 | 123 |
| Mean total varve count | 2466 | 2380 |
| Median varve thickness (mm) | 0.43 | 0.47 |
| Min. varve thickness (mm) | 0.04 | 0.05 |
| Max. varve thickness (mm) | 2.81 | 4.50 |
| Mean varve thickness (mm) | 0.49 | 0.52 |
| Standard deviation varve thickness (mm) | 0.28 | 0.29 |


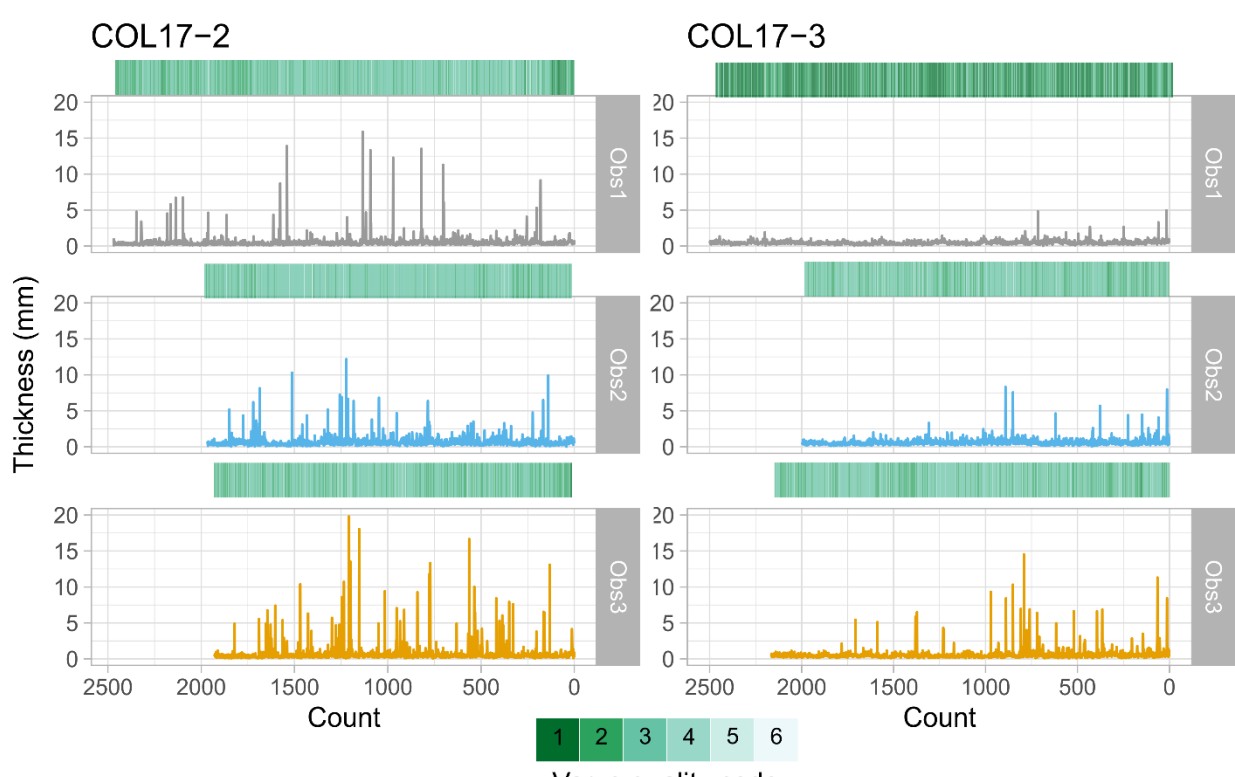

**Figure 6. Observer measurements of varve thicknesses (lines) and quality (heatmaps) for cores COL17-2 and COL17-3.**





**Table 2. Comparison of observer and core-specific varve ages based on symmetric and asymmetric varveR as well as the integrated model. HDR = highest probability density region.**

| | COL17-2 | | | | COL17-3 | | | |
|---|---|---|---|---|---|---|---|---|
| *Symmetrical varveR* | Obs 1 | Obs 2 | Obs 3 | **Average** | Obs 1 | Obs 2 | Obs 3 | **Average** |
| Ensemble mean total count (varve years) | 2749 | 2171 | 2478 | **2466** | 2616 | 2103 | 2419 | **2380** |
| HDR (2.5-97.5%) | 2614-2911 | 2037-2320 | 2351-2617 | **2033-2847** | 2498-2739 | 1958-2249 | 2283-2543 | **1999-2710** |
| Difference from average (%) | +10.9 | -12.7 | +0.5 | **23.6\*** | +9.4 | -12.4 | +1.6 | **21.8\*** |
| *Asymmetrical varveR* | | | | | | | | |
| Ensemble mean total count (varve years) | 3107 | 2590 | 2898 | **2865** | 2899 | 2506 | 2813 | **2740** |
| HDR (2.5-97.5%) | 2015-4182 | 1233-3733 | 1756-3864 | **1417-3923** | 2161-3717 | 1227-3595 | 1699-3811 | **1394-3742** |
| Difference from average (%) | +8.1 | -10.1 | +1.1 | **18.2\*** | +5.6 | -8.9 | +2.6 | **14.5\*** |
| *Integrated model* | | | | | | | | |
| Ensemble mean total count (varve years) | 3470 | 3309 | 3227 | **3308** | 3095 | 3178 | 3138 | **3137** |
| HDR (2.5-97.5%) | 3098-4075 | 3139-3493 | 3091-3370 | **3091-3970** | 2624-3414 | 3036-3333 | 2968-3309 | **2753-3375** |
| Difference from average (%) | +4.8 | 0 | -2.5 | **7.3\*** | -1.3 | +1.3 | 0 | **2.6\*** |

\* ) Indicates the observer agreement as the range in the percentage difference from the mean



**Figure 7. Comparison of original counts by (a and d) observer 1, (b and e) observer 2, and (c and f) observer 3 for dated core COL17-3. In the top row, the modeled varve counts are shown when using symmetrical (dotted envelop) and asymmetrical (shaded envelop) priors. For the symmetrical uncertainty, the median (dashed line) and the 97.5% (dotted region) high density regions are depicted. For the asymmetrical uncertainty, the median (darkest line), 75 (darkest shaded region), and 97.5% (lightest shaded region) high density regions are depicted. In the bottom row, the integrated varve and radiometric models are shown.**

## 4.4 Observer-related uncertainty

Three observers independently measured the varves of cores COL17-2 and COL17-3 in three separate transects (Table 2). The cumulative uncertainty of each observer to the mean was slightly higher for asymmetrical than symmetrical varveR. The uncertainty varied between 0.5 % (observer 3 COL17-2) and 12.7 % (observer 2 COL17-2). Asymmetrical varveR suggests more under-counting for observers 2 and 3 and more over-counting for observer 1 (Fig. 7). However, segment differences





are both positive and negative for all observers, indicating that systematic bias may not be an issue (Appendix A Table A1). The observer agreement is high for minimum thickness but low for maximum thickness (Appendix A Table A2). Observers

disagreed on the number of indistinct sections, pointing to the subjectivity of varve delineations and confidence levels. Agreement on varve quality between observers is low (Fig. 6), indicating further subjectivity. Sections with thicker varves generally correlate across all observers such as between the varve years of 0-100 and 750-1000 in COL17-3 or between the varve years of 1000-1500 in COL17-2 (Fig. 6).

### 4.5 Marker layer uncertainty

As marker layers were assigned by each observer individually, they may not agree between observers. Thus, the varve count between marker layers, or segment count, in each core indicates a combination of inter-site variability due to the sediment quality and observer judgment (Appendix A Table A1). The largest segment difference was 110 % (172 years) for one observer which cannot be explained by marker layer misidentification alone. Instead, it is indicating that one observer identified more indistinct sections than the other observers for one of the sites.

### 405  4.6 Independent validation

The topmost part of core COL17-3 was dated with two independent radionuclide profiles. The $^{210}$Pb activity in Columbine Lake exhibits a gradual downcore decline that reaches equilibrium around 50 Bq kg$^{-1}$ below 8 cm (Fig. 8a). The age at the base of the radionuclide measurements (12 cm) modeled by conventional methods for CRS and CFCS vary widely (Fig. 8c): CRS reaches 1883 ± 14 CE whereas CFCS comes to 1940 ± 13 CE. In comparison, the Bayesian solution has a wider, but

likely more realistic uncertainty at 12 cm yielding a median age of 1784 CE with a 95 % highest density region of 1866-1679 CE. The $^{137}$Cs activity shows a single peak at 3.25 cm (Figure 4.8B) which we attribute to the 1963 CE fallout from nuclear weapon testing. The peak's depth appears younger by 20 to 30 years in the ages modeled from the lead profile: CRS indicates a year of 1996 CE, for CFCS it is 1998 CE, and 1984 CE for Plum.

A total of six radiocarbon dates were used to model the age profile of Columbine Lake sediment (Table 3). Three dates were previously reported by Arcusa et al. (2019) (UCI 196901, UCI, 190157, and UCI 188317) for a mixture of small insects and plant fragments dated with a calibrated-age uncertainty ranging from 20 to 310 years. One new date was discarded as it returned a modern age (IonPlus 3528). Two more dates (IonPlus 3529 and IonPlus 3530) were measured on a mixture of plant fragments, bark, and aquatic insects due to the paucity of organic material found in the sediment. The uncertainty of the

two new dates ranged from 72 to 76 years. The calibrated basal age at 124.5 cm is 2997 (95.4 % probability: 3073-2888) yr BP.

To verify the annual nature of the couplets in Columbine Lake, we compare the topmost part of the varveR model with symmetrical priors to the $^{137}$Cs chronomarkers and the entire sequence to the radiocarbon profile (Fig. 8c and f). Cesium-137





is used for comparison because of its lower uncertainty, as opposed to the lead age models which are not in close agreement among themselves. The varve count and uncertainty by all three observers show a high agreement with the $^{137}$Cs peak, suggesting the couplets are annual. The whole sequence agrees generally well with the radiocarbon profile, particularly in the top 25 cm. Uncertainty surrounding the varve count increases downcore and the varve counts no longer overlap with the radiocarbon uncertainty to a depth of 60 cm. The basal radiocarbon age is older than the mean age estimated by both

symmetrical and asymmetrical varveR by 600 and 250 years, respectively. The cumulative uncertainty of asymmetrical varveR encompasses the radiocarbon basal age, whereas the symmetrical varveR does not. Counts from observer 1 are systematically closer to the radiocarbon age estimate. The comparison with radiocarbon also serves to identify systematic biases which in the case of Columbine Lake varves tend towards under-counting when using symmetrical priors and possibly over-counting when using asymmetrical priors.


**Figure 8. The chronology of Columbine Lake core COL17-3 based on (a) lead and (b) cesium measurements is seamlessly combined with (c, d) radiocarbon samples using the Bayesian models of Bacon and (e) Plum. Plum performs better than (c) conventional lead models of CFCS and CRS when compared to the 1963 $^{137}$Cs fallout peak. The age-depth models counted by three observers and modeled by varveR with symmetrical priors agree well with the fallout peak indicating the rhythmic laminations are annual. Compared to (f) radiocarbon, underestimations in the varve counts appear to accumulate downcore.**




**Table 3. Uncalibrated and calibrated radiocarbon dates.**

| Lab ID | Depth[a] (cm) | Material | $^{14}$C Age ($^{14}$C yr BP) | Error (± 1sd yr) | From[b] (cal. yr BP) | To[b] (cal. yr BP) |
|---|---|---|---|---|---|---|
| UCI 196901 | 27.5 | Insect wing | 520 | 100 | 671 | 319 |
| UCI 190157 | 46.5 | Bryophyte twig, Daphnia ephippia | 1510 | 310 | 2146 | 790 |
| IonPlus 3527 | 52.5 | Daphnia ephippia, insect armour | 2045 | 69 | 2299 | 1798 |
| IonPlus 3528[c] | 77.75 | Daphnia ephippia, charred twig | -935 | 60 | - | - |
| IonPlus 3529 | 85.75 | Daphnia ephippia, charcoal | 2365 | 72 | 2710 | 2160 |
| IonPlus 3530 | 104.5 | Daphnia ephippia, bark | 2845 | 76 | 3170 | 2777 |
| UCI 188317 | 124.5 | Bryophyte twig, Daphnia ephippia | 2875 | 20 | 3073 | 2888 |

[a] Mid-point depth of 1-cm-thick sample

[b] Two sigma range calibrated with IntCal20 curve

[c] Age not used because returned modern





### 4.7 Varve and radiometric data integrated model

One integrated model was created for each observer. The integrated models updated the prior estimates of the counting uncertainties given the constraints from the independent age model and given each observer's varve thicknesses, varve quality designation, and marker layer identification. The models sampled the probability space for 50,000 iterations and the burn-in occurred rapidly in <100 steps (Appendix A Figure A8). The integrated models result in similar cumulative uncertainty to symmetrical varveR but are much smaller than the uncertainty estimated by asymmetrical varveR (Fig. 7). The integrated models also converge more: the difference in the basal age between observers shrinks to 2.6 %, down from 21.8 % in the symmetrical varveR. The posterior likelihoods of over- and under-counting are larger than the symmetrical priors (Fig. 2 compared to Appendix A Figure A9). They also varied with each varve quality code and with each observer (Appendix A Figure A9). The integrated models were more successful at correcting for over- and under-counting for observers 2 and 3 than observer 1 as seen from the more symmetrical cumulative uncertainty for those observers (Appendix Fig, A8).

Each observer's integrated model was combined into one single integrated model, which hereafter is referred to as the 'integrated model'. The integrated model cumulative age extends by 3137 (3375-2753) varve years or 1120 (1358-736) BCE corresponding to a cumulative uncertainty of -384/+238 years (-13/+7 %) (Table 2). The cumulative mean age is older than symmetrical and asymmetrical varveR and the independent model. However, the HDR encapsulates the mean age of the radiometric mode (Fig. 9b). The greatest deviation between the independent model and the integrated model occurs between 30 and 80 cm depth where indistinct sections are most frequent (Fig. 9b). The cumulative uncertainty in the integrated model is lower than asymmetrical varveR and similar to the symmetrical varveR.

The posterior probabilities of over- and under-counting are higher than the prior expectations for all varve quality codes except for the likelihood of over-counting code 3 (Fig. 9a). The probability of over- and under-counting is similar for varve code 1, with a slight tendency for more under-counting (11 % vs 14 %). Furthermore, the probability of over- and under-counting varve code 2 is the same (41 % vs 40 %). In contrast, the likelihood of over-counting varve code 3 is much smaller than the likelihood of under-counting (10 % vs 88 %). However, the distribution of the likelihood of over-counting is much wider than for other varve quality codes indicating this parameter has the least influence on the iterative improvements made by the Gibbs sampler. More under-counting appears with deeper sediment due to the dominance of poorly preserved sediment identified as varve quality code 3. Similar posterior probabilities resulted from re-running the integrated model with smaller asymmetrical uncertainty.





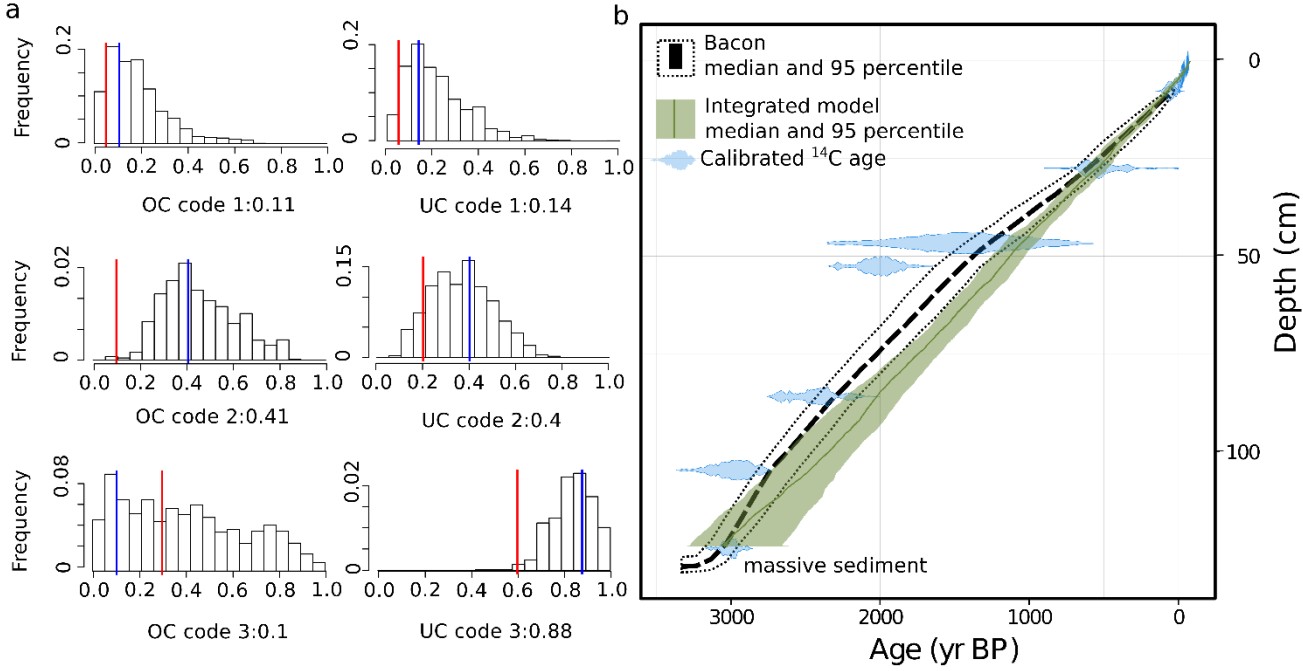


**Figure 9. Integrated varve-radiometric mode. (a) Over- and under-counting posterior distributions for the integrated model for each varve quality code (1, 2, 3). (b) Age-depth model comparison of the independent (Bacon) age model and the integrated model. OC: over-counting. UC: under-counting. Blue line indicates the mode of the posterior distributions. Red line indicates the mode of the prior distributions.**

**4.8 Sedimentation rates**

The estimated sedimentation rate and its uncertainty varied by method and observer (Fig. 10a). Average rates are similar for all varve models with estimates of 0.51 mm/yr (HDR: 0.12-1.45) in symmetrical varveR, 0.44 mm/yr (HDR: 0.08-1.76) in asymmetrical varveR, and 0.42 mm/yr (HDR: 0.08-1.30) in the integrated model. In contrast, rates are doubled on average and higher more frequently in the independent model (0.83 mm/yr, 0.11-3.63) (Fig. 10). The uncertainty range in the varve

models is half of that in the radiometric model.

Sedimentation rates appear more stable throughout the late Holocene in the integrated model than for the radiometric model (Fig. 10b). Periods of higher sedimentation rates occur in the integrated model in the last 100 years, 400-500 BP and 2000-2200 BP. Only the last 100 years of the integrated model shows a similar although subdued trend to the radiometric model.

Furthermore, sedimentation rates are highly sensitive to the observer measurements with relatively little agreement (Appendix A Fig. A10). Those periods of agreement would show lower uncertainty, but uncertainty is relatively stable throughout the record (Fig. 10b).



**Figure 10. Comparison of sedimentation rates. (a) Summary of sedimentation rates calculated with different models
and separated by observer. (b) Late Holocene median (thick lines), 75% (darker shading) and 97.5% (lighter
shading) highest probability density regions estimates of sedimentation rates calculated by the integrated (left) and
radiometric (right) models for the dated core COL17-3. Note the medians of each observer are plotted in the left
panel (thick lines).**



## 5 Discussion

### 5.1 Sources and quantification of uncertainty

Varve chronologies, like all sedimentary profiles, contain uncertainties that stem from complex internal structures, poor quality, technical problems, rapid deposition events, and erosion (Ojala et al., 2012). Unlike other sedimentary chronologies, the errors are propagated by the observer(s) who somewhat subjectively determine what is a varve by "lumping" or "splitting" thicknesses. The sources of uncertainty and their quantification in Columbine Lake are now discussed in turn.

### 5.1.1 Sediment microstructures

The combination of the complex internal structure, shifting structures through time, and thinness of Columbine Lake varves was likely the most important source of uncertainty (Fig. 4b, Sect. 4.2). The complex sub-lamina internal structures the clastic varves are the primary cause of the large uncertainties in observer identification and delineation. It is also likely that laminations are missing due to erosion. Both would result in the under-counting that is particularly evident when comparing 510 the symmetrical and asymmetrical varveR models to the independent chronology (Fig. 8). The systematic bias is corrected by the integrated model. Additionally, uncertainty in the varve delineation impacts the thickness measurements which propagates into the sedimentation rates (Fig. 10). At an average thickness of $0.5 \pm 0.05$ mm, the uncertainty surrounding the delineation of each varve is likely to be proportionately large because of the image quality and pixel resolution used in this study. Missing laminations and misinterpretation due to complex varve structures are common reasons for imprecision 515 (Ojala et al., 2012).

### 5.1.2 Sediment quality

Closely intertwined with the sediment microstructures, sediment quality is likely the second-most important source of uncertainty in the chronology as seen from the prevalence of poor varve quality codes (2 and 3) (Fig. 6). About 78% of the sediment of COL17-2 and COL17-3 was identified as code 2, 3, and 4, all three designations indicating the observer was less 520 than 80 % certain the thickness delineated was accurate. We report a cumulative uncertainty (-13/+7 %) in the integrated model that is on the higher end of values reported in the literature: a cumulative uncertainty of $\pm 1$-3 % is reported in the literature for well-preserved sediment (Ojala et al., 2012) and up to 15 % for unclear, partially disturbed varves in otherwise well-preserved varve sequences (Ojala and Tiljander, 2003; Tian et al., 2005). We also find high estimates of probabilities of over- and under-counting. These uncertainties are not always quantified in the literature, but Ojala and Tiljander (2003) 525 report uncertainties within sections that reach 12 % and indicate more over-counting with depth. Additionally, Fortin et al. (2019) report over- and under-counting estimates of 21.9 and 14.5 %. We find large uncertainty estimates even for the best quality varves in Columbine Lake.





The presence of indistinctly laminated sections was frequently identified in both cores (Fig. 4). The timing of these segments
is generally correlated across both cores, with exceptions, suggesting a combination of macro and micro scale processes. We
accounted for this uncertainty through varve code 5 by emulating varved sediment. Through this analysis, we found that, on
average, more sediment was identified as indistinctly laminated in COL17-2 (25 cm) than COL17-3 (11 cm). In more detail,
the identification and thickness of these segments varied between observers suggesting differences in expert confidence and
indicating high uncertainty may be surrounding the timing of these segments. As a result, the meaning of these indistinct
segments should be interpreted with caution.

### 5.1.3 Technical errors

Technical errors in Columbine Lake varve chronology are likely limited to the sediment embedding and thin-sectioning
process rather than the coring stage. All cores were remarkably similar (Appendix A Fig. A1), and layers could easily be
correlated suggesting the coring process did not disturb the sediment. Although thin sections were overlapped to minimize
sediment loss, the microscopic analysis revealed splits across the sediment in the middle of thin sections likely due to the
embedding process. While infrequent, we accounted for the uncertainty associated with these gaps by using varve code 6,
which tried to quantify the missing sediment. Varve code 6 added an average of 1.2 and 1.7 cm to COL17-3 and COL17-2,
respectively. varveR provides a means of estimating this uncertainty.

### 5.1.3 Rapid depositional events

Errors associated with rapid depositional events were also likely limited to the topmost part of the record. Two thick layers
were found in COL17-2 (1.2-2 and 8.5-9.7 cm) and one in COL17-3 (1.5-2.5 cm). The oldest of the two layers in COL17-2
corresponds to a section of indistinct laminations in COL17-3 (7-8 cm). In situations where one core contains rapid
depositional events, but the other does not, varveR attempts to correct for the missing varves by using information from both
cores. In the case of the oldest layer in COL17-2, only partial information was available from the other core (COL17-3)
because of the indistinct laminations. As a result, information was filled in by the varve emulator which assumed that varves
should be present at that depth. This assumption is likely valid in this case but highlights the emulator should be used with
caution.

### 5.2 Varve formation mechanism

Clastic varves generally form in lakes with (1) favorable catchment properties, lake bathymetry, and hydrology, (2) an
absence of sediment mixing, and (3) a seasonally variable and significant flux of different components (Anderson et al.,
1985; Ojala et al., 2012; Zolitschka et al., 2015). Due to its remarkably clear water and popularity as a backcountry hiking
destination, we could not obtain permitting to instrument Columbine lake to monitor sediment deposition. Therefore, our
understanding of its non-glacial clastic varve formation mechanism is based on field observations, satellite imagery, and
proxy data. Weather data from the region also inform this understanding.




Non-glacial clastic varves form in catchments containing fine-grained silt and clay material, where at least one inflow is present, and where the bathymetry is deep compared to the surface area (Ojala et al., 2000; Zolitschka et al., 2015). In Columbine Lake, most of the catchment (96%) is unvegetated (Arcusa et al., 2019). The production of siliciclastic fine-grained material is likely dominated by the freeze-thaw cycle and hydrolysis. The eroded material is then entrained into the

lake from the margins and via an inflow located to the northwest (Fig. 1). An older inflow is visible from satellite imagery just south of the modern inflow that may also activate during wetter periods. Upon entering the lake, coarser grains settle out first as the energy dissipates and the finer material reaches the coring site. The relatively deep pocket (27 m) where the coring site is located fits the description of a plain sediment depression (Ojala et al., 2000), where depth allows for anoxic conditions and continuous sedimentation and width prevents slope slumping and episodic turbidity currents (O'Sullivan,

570     1983).

The absence of sediment mixing, crucial for varve formation, generally relates to conditions that deter bioturbation (Anderson et al., 1985; Zolitschka et al., 2015). Perennial anoxia, or meromixis, or situations where oxygenation is infrequent enough to deter organism establishment are typical conditions. The deep plain depression where the coring sites

are located likely contributed to this condition, although instrumental data of the water column is unavailable to check for anoxia. Moreover, acidic lake water (pH 5) may be an additional deterrent to benthic biota.

The seasonal sediment transfer in an alpine non-glaciated catchment is usually related to the annual freeze-thaw cycle and runoff events (e.g. snowmelt and rainfall) (Zolitschka et al., 2015). Three types of clastic varves are found in Columbine

Lake with distinct structure, and understanding their formation requires separate mechanisms. Clastic varves are typically composed of a coarse-grained lower and a fine-grained upper lamina produced by a nival discharge followed by winter settling (Zolitschka et al., 2015). Such a progression is found in Columbine Lake's type 1 varves, but not type 2 nor 3. Like previous studies (Cuven et al., 2010), we interpret the type 1 varves silt base (lithozone I) as deposition during the snowmelt season and the clay top (lithozone II) to the settling of fines under ice cover. Two mechanisms can produce the structure of

lithozone I. One possibility is that in the first weeks of snowmelt, the frozen ground and riverbanks inhibit sediment transportation. The resulting stream with low sediment concentration produces overflow conditions. Once in the distal basin, the sediment settles rapidly in ungraded or fining upward sequences (Francus et al., 2008). Alternatively, the initial melt release may occur before the stratification of the lake (Palmer et al., 2019). Either way, we interpret type 1 lithozone I as low-energy, low sediment concentration, nival discharge. The gradual contact between the lithozone I and II may indicate a

slow shut down of turbulence by a slow freeze over and a prolonged period of settling (Desloges, 1994). The sharp contact between lithozone II and the following lithozone I possibly represents the erosive waxing head of the flow, even if the flow is low-energy (Mulder et al., 2001).





In contrast, type 2 varves are composed of subparts that do not follow the typical silt-to-clay progression. The transition from
lithozone I interpreted as low energy nival discharge to lithozone II that is interpreted as winter settling is often interrupted
by lithozone III, a coarse sub-lamina. The combination of lithozone I and III gives the impression of a single sub-laminae
with reverse grading. However, because an erosive contact is sometimes evident between lithozone I and III, this
interpretation is likely ruled out. Instead, a second possible explanation is a high-energy event occurring during or after snow
melt but before freezing over. The San Juan Mountains experience a bimodal precipitation regime with abundant snow in the
winter followed by violent summer thunderstorms. These short-lived summer events may have the energy to transport
coarser material than during the nival snowmelt. As identified in other settings (Cuven et al., 2010), we interpret lithozone
III as discharge events produced by high-intensity rainfall occurring during or after the spring melt.

In varve type 3, lithozone I is replaced with lithozone IV. Rather than a coarse event interruption like lithozone III, lithozone
IV is composed of a single lamina with an inversely graded transition from a dark and fine bottom to light and coarse top
(Fig. 4c). Inversely graded sediment has only rarely been described in lake sediment (Desloges, 1994; Francus et al., 2008;
Guyard et al., 2007; Lewis et al., 2010; Palmer et al., 2019). The primary suggested deposition mechanism is the increasing
underflow velocity of a hyperpycnal flow during the initiation of a flood (Gilli et al., 2013; Lamb and Mohrig, 2009; Mulder
et al., 2001). This depletive (slower velocity with distance) waxing (increasing velocity with time) flow generated by the
steadily increasing discharge (rising limb) at a river mouth (Kneller, 1995) has been attributed to secondary pulses of
sediment in the summer (Desloges, 1994), variable flow from precipitation events (Lewis et al., 2010), and lateral flow of the
sediment to the core site (Palmer et al., 2019). Although possible, this mechanism would require specific discharge rates and
sediment concentrations to produce a current that increases in velocity to a critical discharge rate and is denser than the lake
water in which it enters (Mulder and Syvitski, 1995).


An alternative hypothesis to explain the inversely graded sediment is specific to Columbine Lake and builds on the evidence
of dust-sized sediment in the coarse top of lithozone IV (particle size 5-25 µm). A previous study of Columbine Lake
demonstrated that mineral dust transported from the Southwestern deserts make up 30-57% of the sediment (Arcusa et al.,
2019). As the mode grain size of the dust (22 µm; Neff et al., 2008; Routson et al., 2016) falls within the grain size range of
the top of lithozone IV (Fig. 4c) and dust is regularly found settled on snow in the catchment and on the frozen lake surface
(Appendix A Fig. A11), it is conceivable that the coarse top of lithozone IV would be composed of dust brought into the lake
in one or a combination of three ways. First, dust can accumulate on the lake ice cover and be released as the ice melts, and
this may be later than the onset of snowmelt from the catchment that would create type 1 varves. Second, as snow melts
around the dust particles deposited in the catchment, the concentration of the dust left behind increases (Conway et al., 1996;
Li et al., 2013). A precipitation event late in the nival season could eventually wash the dust into the lake creating the
appearance of inverse grading. Third, the dust-sized material is not dust but catchment material the size of dust brought in
from lake margins or the inlet late or at the peak of the nival season. Varve formation due to aeolian dust has been



documented previously (Zhai et al., 2006), but whether the inversely graded subparts are due to additions of dust and/or hyperpycnites is unclear from the evidence currently available.

**5.3 Varve formation through the Late Holocene**

Three transitions in the varve formation are evident from the stratigraphy and varve analysis. These transitions are either abrupt or gradual and likely reflect important changes in the catchment conditions and/or climate. These transitions will be discussed in turn, from oldest to youngest. Ages and their highest probability density regions (2.5-97.5 %) are indicated from the integrated chronology.


The most abrupt transition in the sequence occurs around 1120 (HDR: 1358-736) BCE (3137, 2753-3375 BP) with the onset of varve formation (Fig. 4) at the contact between units 5 and 4. The processes that can create and sustain conditions necessary for varve formation relate to the physical and chemical properties of the lake water that produce anoxic conditions. These factors include temperature, wind exposure, increased production, and decreased lithogenic influx (Boehrer et al.,
2017; Butz et al., 2017; Makri et al., 2020). In the case of Columbine Lake, lithogenic elements (Ti, Ba, Rb, K) decrease, and Mn/Fe temporarily increases at the transition and is low thereafter (Appendix A Figures A4 and A5). When uncorrelated with detrital elements, as is the case here, high Mn/Fe has been interpreted as high dissolved oxygen concentration in the water column (Naeher et al., 2013). The cause for this momentary increase in oxic conditions is unclear but marks the beginning of the varve formation. Unit 4 directly follows this transition and redox conditions are consistently indicated by
the PCA analysis (Fig. 5b).

The second most evident transition occurs around 60-50 cm depth in COL17-3 (as deep as 72 cm in COL17-2) corresponding to 419-882 C.E. in the integrated varve chronology model with the gradual shift from varve type 1 (unit 3) to 2 (unit 2) (Fig. 4 and Fig. 5). Whereas the asynchronicity of the transition in the cores suggests site specific causes (e.g. processes that oppose varve formation), the fact that both cores eventually transition indicates a catchment wide influence.
The main distinction of varve type 2 is the presence of sub-laminae that are interpreted as higher-energy rainfall events. The position of these laminae within the lamination set suggests the precipitation event occurred late in the nival season, in summer, or in the fall but before the winter settling commenced. The timing of the transition corresponds broadly with the Dark Ages Cold Period (Helama et al., 2017) generally characterized by increased moisture in the southern Rocky
Mountains (Rodysill et al., 2018; Routson et al., 2011) although a period of drought is recorded between 600-700 C.E. at Summitville, 110 km to the south east (Routson et al., 2011). Due to the asynchronicity the timing of the transition cannot be ascertained, and the climatic cause should be interpreted with caution.

The final transition occurs at a depth of 7 cm corresponding to an age of 1874 (1844-1902) C.E. The difference between unit
2 and 1 in the PCA analysis is striking (Fig. 5), with lithogenic inputs distinguishing unit 1 from unit 2. The transition from





unit 2 to 1 appears to occur after the deposition of the deepest massive layer (also a section of indistinct varves in COL17-3). As discussed in section 3.2.2, a large proportion of the sediment appears to be dust-sized sediment (Arcusa et al., 2019). Additionally, the last 150 years coincide with a 1.7-fold increase in dust deposition compared to pre-industrial times in the San Juan Mountains (Routson et al., 2019), with two peaks in deposition occurring around 1880 C.E. and 1950 C.E. as seen

from previous work at Columbine Lake (Arcusa et al., 2019) as well as other lakes in the region (Neff et al., 2008; Routson et al., 2019, 2016). These peaks correspond to the timing of the massive layers: 1973 (1959-1987) C.E. and 1851 (1824-1876) C.E. It is thus conceivable that the additional dust may have disrupted the varve formation process in the massive layer and may have altered the varve formation mechanism subsequently.

A final hypothesis for the transition to varve type 3 relates to an increasing, even if slight, human impact on the catchment as indicated by two structures and other evidence of grazing and mining activity (Fig. 5). Although the catchment is not accessible by road, a rock shelter was constructed on the south shore. The high alpine meadows have been subject to sheep and cattle grazing since the late middle to late 1800s (Baker, 2020) coinciding with the increased lake productivity indicators seen in unit 1 (Fig. 5). The increased productivity and organic content could explain the thicker varves but not the reverse

grading. Secondly, a 2-m-high dam was constructed at the outlet for Mill Creek presumably sometime around the turn of the 20th century, to raise the lake water level and secure water rights for a downstream mine (pers. comm., Forest Service at San Juan National Forest). This water level increase and fluctuation could have increased erosion and reworking of hillslope sediment. Finally, mining became increasingly prevalent in the area from the 1800s (Blair and Bracksieck, 2011), although we did not find evidence for mining within the catchment. Mining indicators (e.g. Guyard et al., 2007) such as silver and

zinc become abundant in unit 1, and the increase in heavy metals could have changed both lake productivity and signal a change in lithogenic input. Whether the unique varve type 3 reflects the input of dust or sediment from shoreline or hillslope sources, or changes in lake productivity, or all these factors together, the change occurs in the industrial period and is likely related to human activities within and beyond the catchment.

## 5.4 Integrating varves with radiometry

Radiometric ($^{14}$C, $^{210}$Pb, $^{137}$Cs) profiles are frequently used to validate varve chronologies (Ojala et al., 2012; Zolitschka et al., 2015); however, ages derived from radiometric profiles are generally systematically older than the varve chronology for various reasons (Bonk et al., 2015; Tian et al., 2005; Żarczyński et al., 2018). As the varveR output for Columbine Lake consistently shows this divergence (Fig. 8f) we now discuss the merits and pitfalls of integrating the varve chronology with the independent radiometric age-depth model by exploring three possibilities: (1) the varveR model is accurate and the

calibrated $^{14}$C dates are older than the true sediment ages; (2) the calibrated $^{14}$C dates are accurate and the varveR model underestimates the true sediment ages; or (3) both the model and the calibrated $^{14}$C dates have unknown systematic biases.





Radiocarbon dating in high-elevation lake sediments is often challenged by a paucity of adequate organic material (e.g. Arcusa et al., 2020; Schneider et al., 2018). To gather enough material for a standard graphite-based AMS measurement, the

radiocarbon samples in this study were composed of a mixture of aquatic and terrestrial material (Table 3). Samples of mixed composition have been shown to yield ages that are generally too old (Zander et al., 2019). Both aquatic and terrestrial macrofossils are associated with processes that can increase their apparent age. For example, aquatic organisms are subject to a hardwater effect due to dissolved inorganic carbon synthetization (Geyh et al., 1998, 1999), whereas terrestrial material might be significantly older than the enclosing sediment because of the lags between growth and deposition (Bonk

et al., 2015). At least one of the seven radiocarbon dates is likely too old (IonPlus 3527), exceeding Bacon's 95 % uncertainty band (Fig. 8f). A leave-one-out cross-validation analysis (e.g. Parnell et al., 2011) could help identify other outliers but the analysis was not undertaken in this study. Despite the potential for other samples being too old, the integrated chronology overlaps with all other radiocarbon samples (Fig. 9b), and the divergence between symmetrical varveR and the radiometric independent model appear to increase with depth (Fig. 8f), both of which support the accuracy of the varve-

based age model.

A younger varve chronology compared to the independent model would indicate varve under-counting. Varve count underestimation is recognized in sediment with poor varve appearance (Tian et al., 2005) and depending on the method used in building the chronology (Żarczyński et al., 2018). As discussed in section 5.1, both the sediment microstructures and the

quality of the varve appearance are important sources of uncertainty in Columbine Lake: varves are thin, complex, and their formation mechanism appears to change through time. Additionally, the varve emulator is unlikely to have over-estimated the varve counts given the relatively stable sedimentation rate through time. Although observer bias does not appear important, since age deviations from the mean are both positive and negative, and for the reasons listed above, it is most likely that systematic under-counting is prevalent. The integrated model satisfies all available evidence and is more accurate

than relying on a single chronological method.

## 6 Conclusion

A multi-core, multi-observer varve chronology extending 3137 (-13/+7 %) years was produced from thin and complex varves from high-elevation Columbine Lake, Colorado. A varve formation model was proposed and was demonstrated to shift through time most likely due to climate in pre-industrial times and human influence within the catchment and on

regional dust emissions in industrial times. A Bayesian model was used to quantify the uncertainty associated with the quality of the varve appearance, the indistinct and intermittent varves, technical issues, observer judgement and depositional events. The varve chronology was integrated with an independent radiometric ($^{14}$C, $^{210}$Pb, and $^{137}$Cs) age-depth model to estimate the probabilities of over- and under-counting for different varve quality codes, reduce cumulative uncertainty, and correct for systematic under-counting.






This approach to building a varve chronology goes beyond the estimation of age uncertainty as it also constrains the uncertainty around varve thickness and thus sedimentation rates. The integration produced estimates of sedimentation rate that combine short-term as well as some long-term information, native to the varve and the radiometric chronologies. Furthermore, the approach offers an ensemble of plausible sedimentation rates from which flux and its uncertainty can be

calculated. This work not only establishes the chronology and sedimentation rates of Columbine Lake sediment to anchor future research at the site, it also demonstrates the potential for expanding high resolution reconstructions even to sites with indistinct and intermittent varves.





## 7 Appendix A

**Table A1. Difference in counts between marker layers between cores for each observer. Note that marker layers do** 735 **not cross-coordinate between observers, only between cores for each observer. Difference is calculated as COL172-COL17-3.**

| Marker Layer | COL172 | COL173 | Observer | Difference (years) | Difference (%) |
|---:|---:|---:|---:|---:|---:|
| 1 | 699 | 660 | 1 | 39 | 5.7 |
| 2 | 275 | 308 | 1 | -33 | -11.3 |
| 3 | 951 | 1230 | 1 | -279 | -25.6 |
| 4 | 439 | 321 | 1 | 118 | 31.1 |
| 5 | 9 | 8 | 2 | 1 | 11.8 |
| 6 | 124 | 74 | 2 | 50 | 50.5 |
| 7 | 214 | 187 | 2 | 27 | 13.5 |
| 8 | 41 | 91 | 2 | -50 | -75.8 |
| 9 | 203 | 165 | 2 | 38 | 20.7 |
| 10 | 442 | 411 | 2 | 31 | 7.3 |
| 11 | 180 | 271 | 2 | -91 | -40.4 |
| 12 | 69 | 182 | 2 | -113 | -90.0 |
| 13 | 206 | 221 | 2 | -15 | -7.0 |
| 14 | 252 | 192 | 2 | 60 | 27.0 |
| 15 | 128 | 145 | 2 | -17 | -12.5 |
| 16 | 9 | 7 | 3 | 2 | 25.0 |
| 17 | 34 | 25 | 3 | 9 | 30.5 |
| 18 | 46 | 30 | 3 | 16 | 42.1 |
| 19 | 56 | 21 | 3 | 35 | 90.9 |
| 20 | 212 | 177 | 3 | 35 | 18.0 |
| 21 | 43 | 99 | 3 | -56 | -78.9 |
| 22 | 185 | 169 | 3 | 16 | 9.0 |
| 23 | 240 | 256 | 3 | -16 | -6.5 |
| 24 | 148 | 115 | 3 | 33 | 25.1 |
| 25 | 59 | 70 | 3 | -11 | -17.1 |
| 26 | 183 | 266 | 3 | -83 | -37.0 |
| 27 | 70 | 242 | 3 | -172 | -110.3 |
| 28 | 80 | 156 | 3 | -76 | -64.4 |
| 29 | 106 | 155 | 3 | -49 | -37.5 |
| 30 | 212 | 193 | 3 | 19 | 9.4 |
| 31 | 176 | 139 | 3 | 37 | 23.5 |





**Table A2. Observer- and core-specific varve sequence statistics of thickness and counts. Varve quality codes 4, 5, and 6 are excluded from the analysis except to calculate the cumulative length of indistinct sections. All units are millimetres unless otherwise noted.**

| Core name | COL17-2 | | | COL17-3 | | |
|---|---|---|---|---|---|---|
| | Obs 1 | Obs 2 | Obs 3 | Obs 1 | Obs 2 | Obs 3 |
| Minimum thickness | 0.05 | 0.01 | 0.07 | 0.03 | 0.02 | 0.1 |
| Maximum thickness | 2.32 | 3.64 | 2.46 | 4.94 | 1.69 | 6.86 |
| Median thickness | 0.39 | 0.48 | 0.41 | 0.44 | 0.51 | 0.46 |
| Mean thickness | 0.43 | 0.56 | 0.48 | 0.48 | 0.56 | 0.51 |
| SD thickness | 0.23 | 0.35 | 0.26 | 0.24 | 0.25 | 0.37 |
| Total indistinct section length | 40 | 10 | 108 | 167 | 57 | 112 |


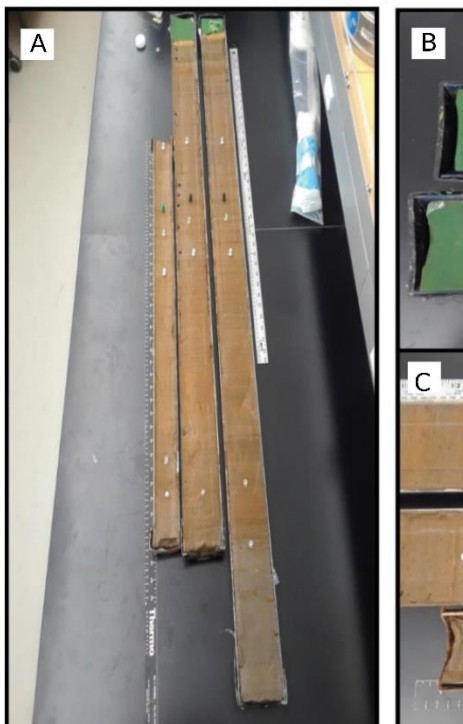

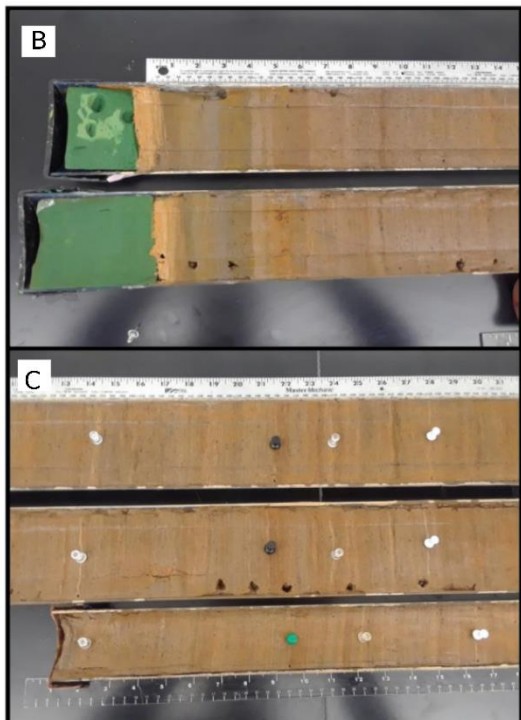

**Fig. A1. Tie points from three Columbine Lake cores. (A) shows COL-17-2 shown on the far right, COL-17-3 in the middle, and COL-16-1 on the left. The top of cores COL-17-3 and COL-17-2 are shown in (B). (C) is a section of the middle of all three cores with matching laminations marked with pins. Image credit: Wiman, C. (2019). Late Holocene hydroclimate and productivity in varved sediment at Columbine Lake, Colorado (Master thesis, Northern Arizona University).**





Fig. A2. Examples of varves appearance for each varve code.



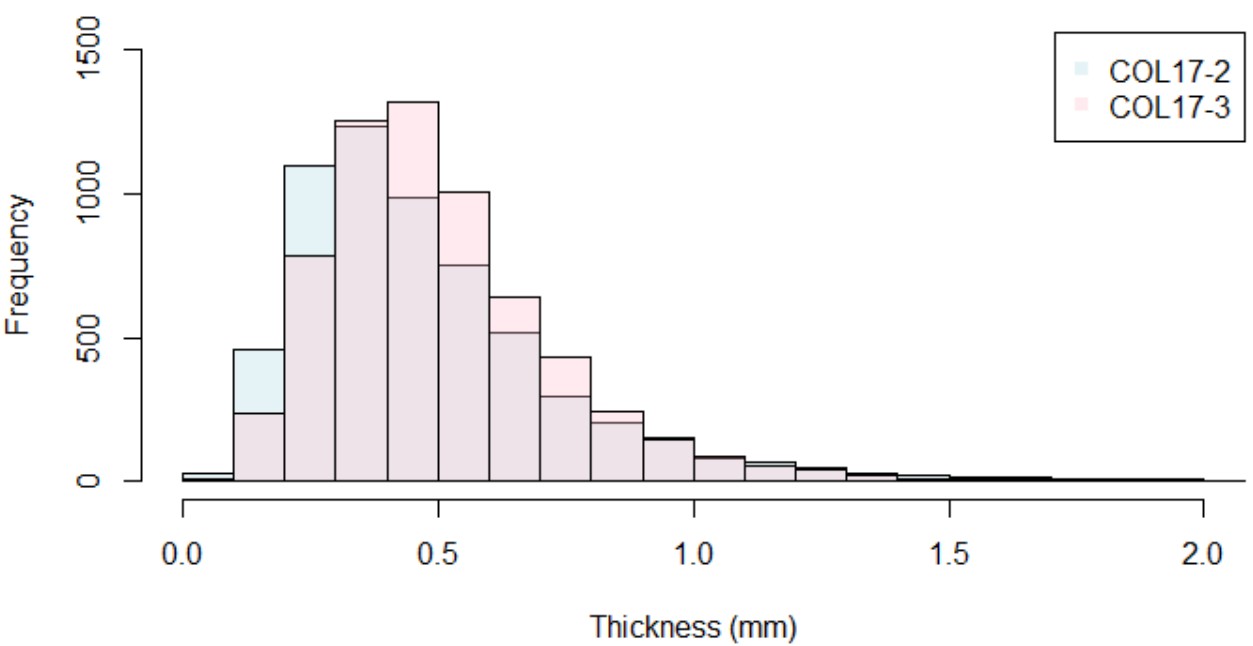


**Fig. A3. Comparison of varve thicknesses from varved sections (codes 1, 2, and 3) between COL17-2 and COL17-3.**

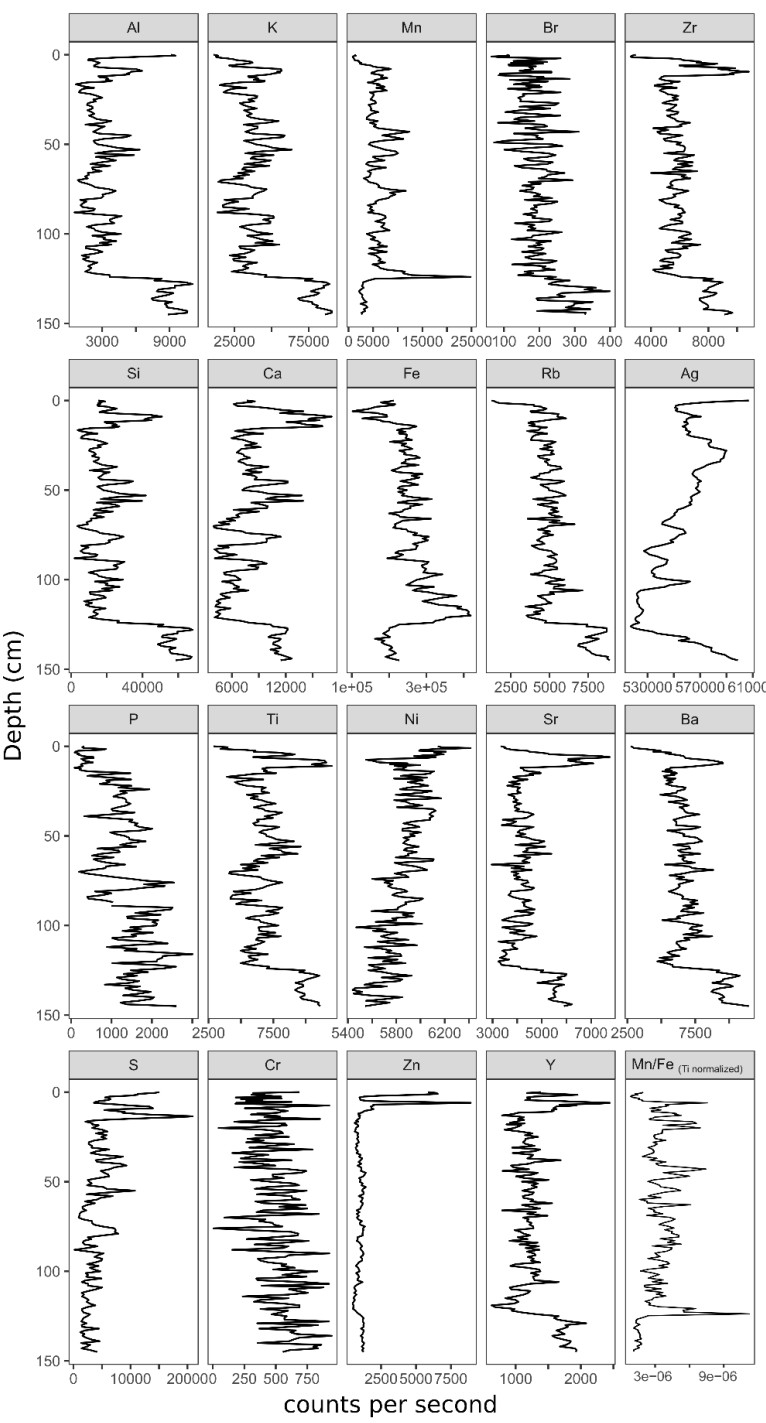

**Fig. A4. X-Ray Fluorescence (XRF) elements measured on core COL17-3. Ratio Mn/Fe is normalized to Ti counts.**



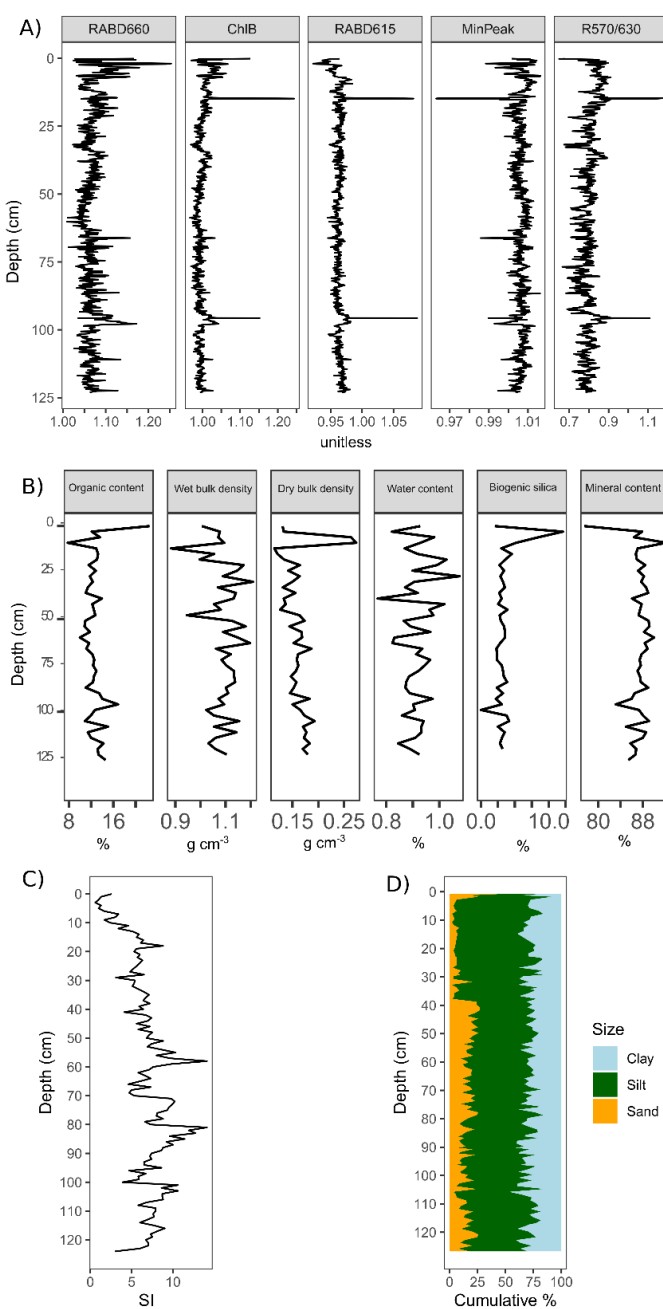

**Fig. A5. Proxy dataset measured on core COL17-3 and used for the statistical analysis. (A) Ratios RABD660, RABD615 and ChlB are indicators of productivity. Ratios minPeak and R570/630 are indicators of clay minerals. (C) Destructive analyses including LOI (organic, water and mineral content), biogenic silica and wet and dry bulk density. (C) Magnetic susceptibility. (D) Grain size.**






**Fig. A6. Spearman's Rank correlation plot.**





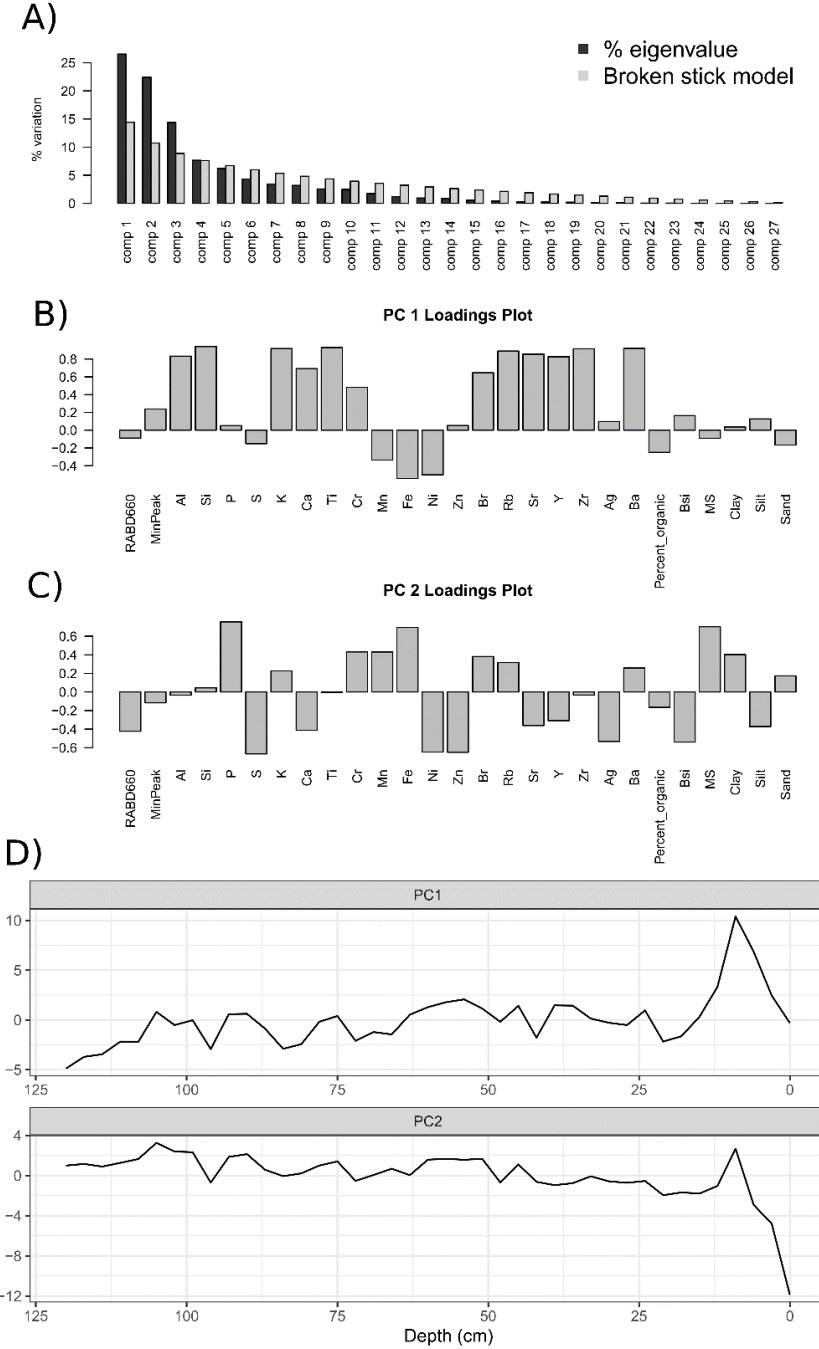

**Fig. A7. (A) Broken stick model used to identify the significant principal components. Although three components are significant and explain 63.3 % of the variability, only components 1 and 2 are discussed in the manuscript. (B-C)**

**Loadings and (D) scores for components 1 and 2.**





**Fig. A8. Integrated model diagnostics. Objective function output value (left) and counting probabilities (right) for each iteration for observers 1 (top), 2 (middle), 3 (bottom). OC = over-counting. UC = under-counting. Number that**
**follows OC/UC indicates the varve quality code.**







**Fig. A9. Posterior probabilities of over- and under-counting for each observer for core COL17-3. Comparison between independent and integrated age-depth model. OC: over-counting. UC: under-counting. Code 1-3 indicate the**
**varve quality codes 1, 2, 3.**



**Fig. A10.** Sedimentation rates for each observer for symmetrical varveR, asymmetrical varveR and the integrated models.



A) June 2012

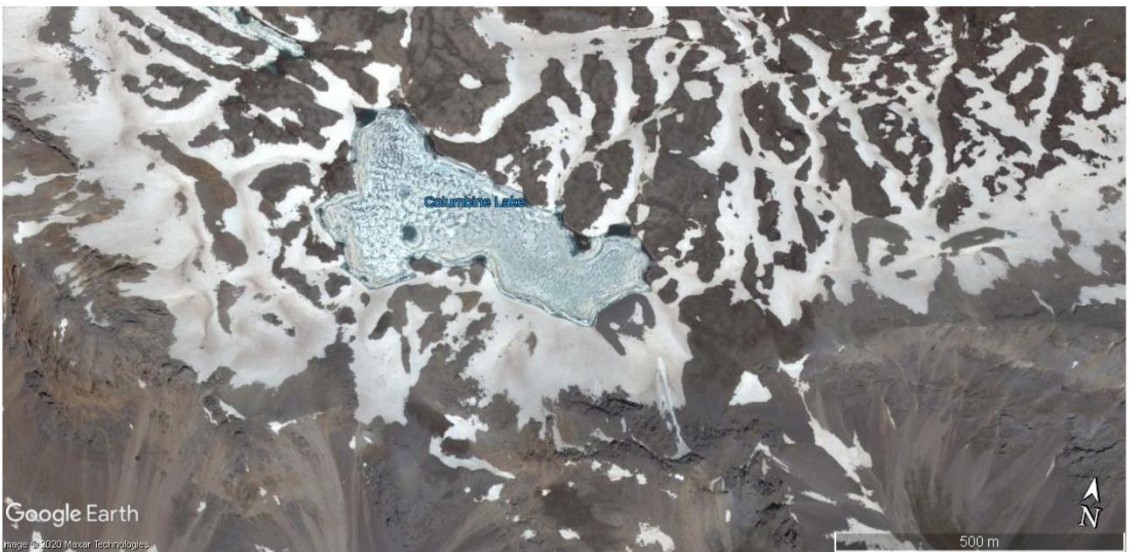

B) June 2014

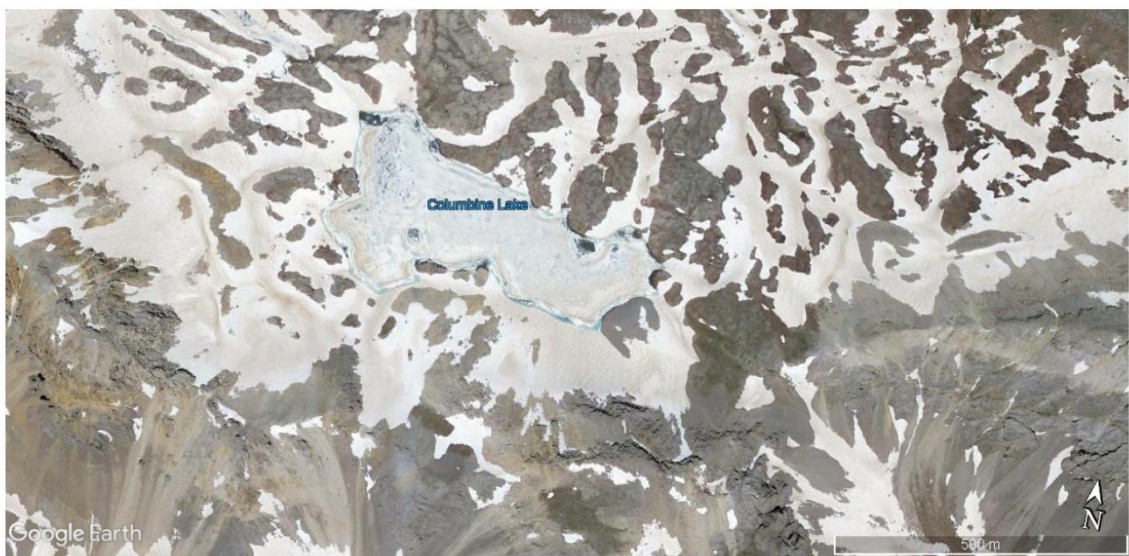

**Fig. A11. Dust deposition in Columbine Lake catchment and frozen surface is frequently visible as a light brown tint on the snow. Image from (a) June 2012, and (b) June 2014. Map data: © Google Earth.**


**8 Code and data availability**



Code for the original VarveR model can be found at 10.5281/zenodo.4733326. Code for the varve and radiometric model integration can be found at 10.5281/zenodo.4744872. Datasets containing radiometric measurements from Columbine Lake

can be found at https://doi.org/10.25384/SAGE.9879209.v1. Datasets of varve delineations can be found at 10.6084/m9.figshare.14251400. Datasets necessary to run the code (LiPD file and Bacon output file) can be found at 10.6084/m9.figshare.14417999. Dataset containing proxy measurements produced in this study can be found at 10.6084/m9.figshare.14265644 for the review process and will be uploaded to NOAA World Data Service for Paleoclimatology upon publication.

**9 Author contribution**

SHA and NPM conceptualized the study. CW sampled and embedded the sediments, SHA, CW, and SP measured varves, and SEM measured lead samples. MAL ran the Plum-Bacon model. SHA and NPM created and modified the Bayesian models, and SHA ran the models. SHA visualized the data and drafted the original manuscript. All authors contributed to the review and editing.

**10 Competing interests**

The authors declare that they have no conflict of interest.

**11 Acknowledgments**

This research was funded by Bob and Judi Braudy and we are grateful for their support. MAL was partially founded by CONACYT CB-2016-01-284451 and COVID19 312772 grants and a RDCOMM grant. We thank D Buscombe for letting us

use the bathymetric equipment, RS Anderson for the identification of macrofossils for [14]C dating, K Whitacre for lab assistance, Rosalind Wu from the San Juan National Forest Service for working with us to obtain permits for sampling Columbine Lake, C Routson and D Kaufman for helpful feedback on the manuscript, Quality Thin Sections for producing the thin sections, and C Ebert for conducting [14]C dating at IonPlus in Zurich. We thank E Yackulic, C Mogen, W Wiman, C Routson, A Wong, A Platt, J Chaffeur, E Broadman, and M Caron for the help in the field.

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
