# Peer review of "A Bayesian approach to integrating radiometric dating and varve measurements in intermittently varved sediment"

_Geochronology, 2021_

## Author Response (AR1)

**Point by point changes and responses to reviewers**

**Main changes**

- At the request of both reviewers, the geochemical and paleoenvironmental data as well as related text and figures were removed to focus the manuscript solely on the new approach. The manuscript now explores the new approach and demonstrates it on Columbine Lake, without delving into the paleoenvironmental aspects of the record.
- The text was generally shortened, which greatly improved readability.
- In the text we made clearer the reasoning for pursuing low quality varve records and the broader implications of having an approach that can extract information from such records.
- We added explanation of why the Cs peak is unlikely to be Chernobyl.
- We clarified why the parallel cores do meet the requirements of reproducibility.
- We clarified in the text that our method does not "count" varves in the traditional sense of varve "counting".
- The code was reworked to be easier to run, not require the user to change the paths, and some necessary custom functions were added that were previously omitted by mistake.

**Anonymous Referee #1**

Comment 1: I suggest that the Authors reconsider the title. Given that the manuscript shows a promise for more general remarks it should be the focal point of the title. It looks like a great tool for sediment profiles with missing varves, but for entirely varved sequences it can be even better. There are strong points in the paper: R package, Bayesian varve-chronology model, integration with radiometric dates and so on.

We changed the title to "A Bayesian approach to integrating radiometric dating and varve measurements in intermittently varved sediment".

Comment 2: This paper is introducing a new methodological concept, yet it lacks clearly resolved suggestions at the end (think Fig. 3 but even more general), and outlook. Generally speaking – which codebase and repository should be used in the future? One model integrates Gibbs sampler, the other does not and so on, is it going to be merged? By the end of the paper, there may be some expectations from the potential readers, that are not met at this point.

We added the expected outcomes of the paper in the conclusion and clarified in section 8 that we are willing to work with interested parties to make the code work.

Comment 3: Integration of the geochemical data. It is interesting and useful to describe sediments, varve types and their formation (it is used to cluster the data, for example) and overall paleoenvironmental conditions. Yet, it in my opinion, especially given the Introduction there is no clear indication for the geochemical data importance, it was unexpected addition. Because of that, manuscript seems to trail a little bit out of focus and is discussed with details that are not necessarily important. It

is only at the end of the paper that the geochemical data is brought to the attention. There it is partially justified but it lacks in the context of the chronology development, even though it provides insights into the factors influencing the varve quality.

We removed the geochemical data and paleoenvironmental discussion from the text.

Comment 4: Authors suggest that missing lamina can be a result of erosion. Is there any evidence that over the history changing productivity and mixing could have changed formation and preservation of varves more than erosion? (5.1.1)

This was no longer relevant as the paper no longer discusses how the varves were formed.

Comment 5: Do the Authors see possibility to expand the model with, for example, results from varve counting on geochemical and geophysical maps/profiles, which sometimes show better rhythm and separation of varves. If so, how varve quality codes would be associated with that kind of data?

We added text on the potential of the method using other types of data on line 624.

Comment 6: Do the Authors see possibility to employ at least some of the proposed advantages to one-core sites? If so, then include it in the text. Multi-core research is often cost and time consuming, and not always possible.

We added to text on line 624 that the method could be amended for one-core sites.

Comment 7: I think that for the discussion of the rationale, there should be reference to VARDA database (Ramisch et al., 2020; https://doi.org/10.5194/essd-12-2311-2020) – an attempt to standardize varve chronologies with Bayesian modeling – it shows that it is an important issue worldwide.

This was added to line 82.

**Abstract:**

L10-11: weather alternative archives are available or not varves still provide superior archives.

We added text to say that even varve archives with indistinct varves are still highly valuable to lines 56-58.

L11: chronology goes before paleoenvironment.

This was changed.

L15: "sediments", thin sections are just mean to preserve and investigate the material.

This was revised.

**Introduction:**

L35: maybe: due to time, density, and supply – however please consider that density itself is affected by time (compaction, post-depositional processes) so I'm not sure about putting it on the same line as time and source. In general, fluxes seem to be better but once there is a drastic change in sediment accumulation rate most of the constituents will likely follow. So relative representation is also useful.

We clarified that both are useful.

L68-9: I'm not sure if parenthesis information adds to the introduction.

This was removed.

L73-4: "For example" at the beginning of the sentence and at the end.

This was changed.

L75: reveal intervals where missing varves can be inserted.

This was changed.

L80: I'm missing a link between these two paragraphs.

We tried to improve.

Last paragraph: Consider changing the order, first introduce rationale – there are few robust, long chronologies in the area and so on, therefore Columbine provides somewhat unique research material.

This was changed as suggested.

**Study site:**

L89/103: It' a matter of preference but maybe add m.a.s.l.

This was changed.

L91: formation and preservation…

This was changed.

L103: rephrase maybe so climate is not typified by the climate itself.

This was changed.

**Methods:**

L114: Probably UWITEC, without "H".

This was changed.

L117: "Error! Reference source not found"

This was fixed.

L118: shallower depth meaning? Please put coring depth close to the respective core codes in L114. It's interesting to see what the change in depth was making that much difference in varve preservation.

This was updated.

L121: analyzed with non-destructive rather than for.

This was changed.

L128: minimum peak of what? Minimum reflectance peak/Rmean?

This was no longer relevant as the paper no longer uses that data.

L134: what was the treatment?

This was no longer relevant as the paper no longer uses that data.

L135: you provided details for previous methods, what was the equipment used here?

This was no longer relevant as the paper no longer uses that data.

L148/155: reference to the papers is fine, but maybe consider adding package versions of rbacon, rplum and SERAC?

All versions were added.

L150: line is redundant: statistical framework that uses statistical inference.

This was removed.

L151: why do you mention CRS particularly in this place?

This was amended. CRS was mentioned to explain that Plum does a better job at estimating uncertainties than CRS, which is one of the most common models used.

L165: calibrated for/to what?

This was no longer relevant as the paper no longer uses that data.

L175: what R version? There are important changes between the big versions like 3, 4 or 4.1.

Version was added.

L181: curious: why FactoMineR rather than base PCA?

For esthetics!

L194: principle is general, so sections rather than thin sections.
This was changed.

L196: do you consider developing approach based on other file system, rather than proprietary shapefiles?

Added explanation to line 168.

L196-7: what is necessary to record: depth and code/name of layer?

Made this clearer on lines 165-167.

L215: if you introduce the paragraph properly, I see no need for 220-227 to be discussed after the code 6. It can and should be discussed immediately after 4. Especially that you introduce idea of simulation as code 6 was "similarly" emulated, but it is presented afterwards.

We tried to restructure.

Paragraph starting at L229 needs to be more concise.

We tried to be more concise.

L245: there is already subject in the sentence, no need for Columbine Lake at the end.

This was removed.

L248: and expert judgment.

This was changed.

Paragraph at L255: rewrite to avoid redundancy from previous sections in Methods and try to be more concise. L260 no need for parenthesis.

We tried to be more concise.

3.8 and 3.9 can be as well one section. This subdivision seems unnecessary. Avoid redundancies.

Also here, we tried to restructure.

L272: that kind of detail on algorithm etc. could go into the code comments or other kind of supplementary materials.

Instead of moving all the details to the supplement, we reworded to be more concise because we thought there was not enough material to move into its own supplemental section. We can amend this if it still seems like too much detail.

L277: if by adjusting – typo?

Yes, typo was fixed.

**Results:**

L303: I don't think that "redox" should be used as an adjective for any element (Mn). Its state is controlled/sensitive to redox changes.

This was no longer relevant as the paper no longer uses that data.

L313: in the microfacies analysis…

This was no longer relevant as the paper no longer uses that data.

Section 4.2 Please indicate that further in the text you are using "lithozones" to describe seasonal/event layers and structures within the varves. Also, first and second sentence can be shortened and merged into one.

To meet reviewer 2 comments, we changed "lithozones" to "assemblages.

L353: space missing in <5 (or remove space in previous notations).

No longer relevant.

L356: deeper. Try to be more concise.

This was changed

L410: while I get that Bayesian model must produce confidence interval, the density region starting at 1679 CE seems to be strange by the definition if it is on the depth where unsupported Pb just vanished.

We agree. We added text to line 413 to clarify this bug.

L415-420: maybe move some of the details to Methods and avoid repetition.

We tried to be more concise.

L450: space missing in <100 (or remove space in previous notations).

This was changed in every instance.

L4620: missing "l" in model.

This was fixed.

**Discussion:**

L501: First part of the sentence and subjects are in mismatch. Chronologies versus profiles.

This was amended.

L507: structures of the…

This was amended.

L540: this is rather typical. Often these splits will occur on the unconformity between the layers of different density and properties.

It was clarified that we added this detail to show that the method can also try to estimate the uncertainty due to these splits.

Paragraphs from L560 to 590: although discussion of anoxia is important it is inserted between the two paragraphs on the source of the material and seasonal differences of supply. Furthermore, Authors repeat some of the information. Please, consider reworking this section.

This was no longer relevant.

Paragraph from L635: I advise caution in the Mn/Fe interpretation, even if these are mostly uncorrelated to lithogenic inputs. Seasonal relations are of importance.

This was no longer relevant.

Paragraph from L645: what about sediment focusing? Also, it seems that you started writing "C.E." from here on rather than previously used "CE".

This was amended to be consistent throughout.

**Figures:**

Figure 1: Color for vegetation is virtually the same as depth of 0 meters of the lake. Outline the lake and catchment, so legend/key and scale are separate entities. Elevation – see comment to L89. Inflow line in the key and on the map is narrow and hard to see. Caption: Columbine Lake.

We changed the figure to use a different color key. The caption was changed. The lines were thickened. It was not clear what was meant by the comment about the outline.

Figure 4: Reconsider legend/key placement and structure. For example, varve types corresponding to the color bars in (a) are at the bottom of the box, and easy to omit.

We changed the figure.

Figure 5: Is it correct to describe the dendrogram as constrained or not, or rather it is a dendrogram showing the results of constrained clustering method? (b) increase contrast of the vector groups?

This was no longer relevant.
Figure 7: Figure shows an impressive improvement of the chronologies once they are integrated with the results of the radiometric dating methods. Yet, this part is discussed later in the paper, with figures in between.

This figure should now be more integrated with the text, with the geochemical text having been removed.

Figure 8: Some of the figure caption is a repetition of the text, rather than figure description.

The caption was amended.

Figure 9: missing "l" in model.

This was fixed.

Figure 10: Any comment on multimodal distribution?

We added text to comment on this on lines 496-499.

Figure A2: Consider adding depth/length scales.

We were not able to do so, so we changed the caption to explain this is only for example.

Figure A4: How was the Mn/Fe ratio normalized?

This was no longer relevant.

Figure A6: Matter of preference, but warmer color is associated rather with positive values and colder with negative, regarding correlation.

This was no longer relevant.

Please check figures and their captions in the text and Appendix for consistent use of (A), (B) and so on. For example, Fig A1, A5, A7 – capitalized, whereas others are not.

This was amended throughout.

**Code:**

Please think about very short readme file describing the actual workflow and data structure to reproduce your research.

A readme file was added.

It seems that in the "varveR_Gibbs-v.1.0.0" there are missing extensions in the R scripts, which I assumed for the review to be typical ".R" files.

This was a mistake during the upload and all files were changed.

If the Authors consider releasing the same scripts with the Manuscript please check the code for instances like below, where full paths are provided.

readLipd("D:/OneDrive for Business/…"); I cut the remaining path for clarity. Consider using here::here() or likewise so code is more reproducible. After this point I did not change all the paths to run the code.

The code was amended so this is no longer an issue.

Overall, code is commented, and consecutive blocks are explained. Examples are provided for functions and parameters are described. For the future releases of varveR some code cleanup is necessary, though.

**Anonymous Referee #2**

48: The sentence "Error sources are associated with (1) inter-site differences in varve counts…" needs to be corrected to "intra-site differences", inter-site differences make no sense here as the site is the lake. The same for 185 and 195.

This was amended.

88ff: Fig. 1a displays some evidences for a delta south of the inflow currently not in use. Furthermore, there is another lake basin in a distance of only ca. 150 m west of Columbine Lake, which is probably acting as a sediment trap for coarser sediment fractions before they enter the studied lake. All this needs to be mentioned and discussed and might have implications for interpretations.

The presence of the inflow and possible delta was made clearer on line 94.

122-124: "(2 cm measurement diameter resolution)" – please reword, this is difficult to understand.

This is no longer relevant.

171ff: "Therefore, we used point counts and length measurements directly on individual grains in the slides. At least 100 grains were measured from the varve transects." Please explain this procedure with other words. As it is now, I am not understanding what has been done.

This is no longer relevant.

184ff: in the chapter "Description of the original varve model" the lamination is neither described nor confirmed as varved. Throughout the entire manuscript the presence of varves is regarded as prior but unproven information.

We noted the misunderstanding with the title "Description of the original varve model" which contrary to the Reviewer's comment does not refer to Columbine Lake but to the VarveR algorithm (i.e., the model) presented by McKay (2019). In the revised manuscript we made this distinction clearer on lines 146-148.

300: Characterization of unit 5 by the grain size clay is not supported by the data (as shown in Fig. 4).

This is no longer relevant.

301: Fig. A5 does not show data of unit 5; the same is probably true for Fig. A6. Furthermore, it remains unclear, which data is shown in the correlation matrix.

This is no longer relevant.

303-304: This sentence is true for Fe but not for P. Moreover, the drastic decrease of siliciclastic elements needs to be mentioned as well.

This is no longer relevant.

323f: The sentence "Some heavy metals (Zn, Ag) also increase to their maximum levels (Appendix A Fig. A4)." is only partly true and questions the interpretation (cf. 679), as Ag has similarly high values at the base of the record.

This is no longer relevant.

352: Why is the mineralogy provided only for type 2? However, this data is not used for any interpretation, it may as well be deleted.

This was removed.

364: Std. deviation provided in the text is 0.05 mm and distinctly different from the one provided in Tab. 1 (0.3 mm).

We checked this. The text and the table values were calculated differently which caused the discrepancy. We changed the text so it is now consistent with the table.

389: "Three observers independently measured the varves…" Here it is necessary to name those who counted (not only in the chapter "Author contributions")! Are these three experienced sedimentologists or students? Additionally, it is not explained how the varves were counted.

The contributions of each author are described in the Author Contribution section, per journal policy. We added a sentence to line 174.

*Regarding how the varves were "counted": on original line 195 we stated "In this study, thickness delineations were created as ArcMap shapefiles". In the new manuscript we amended the text on line 151.*

400: Since DeGeer, marker layers are assigned macroscopically (in the case microfacies analysis is applied, this can be extended to microscopic marker layers) to distinct layers or changes in sediment composition to ease the counting of shorter sections of a profile (between individual marker layers) by different observers. I do not understand, why every observer sets up his or her individual set of marker layers in this study.

*We did assign marker layers macroscopically (Figure 1) as we all as for each thin section. Each observer set up their own marker layers so the model could include an estimate of marker layer uncertainty as well, an important additional source of uncertainty. We made this clearer in Table A1.*

Before 500: the "Results" chapter very marginally describes and discusses sedimentological and geochemical data as they are shown with Figs. 4a, A4 and A5. Thus, the question arises, why this manuscript is expanded largely by including such data?

*We removed the geochemical data.*

564: At the high altitude of the lake, "hydrolysis" (chemical weathering) is certainly of very little importance if at all.

*As we removed the paleoenvironmental discussion, this is no longer relevant.*

568-569: Please explain the reasons for anoxia to develop in an oligotrophic alpine lake with clastic sediments like Columbine Lake?

*As we removed the paleoenvironmental discussion, this is no longer relevant.*

644-645: The argument provided here and directly linked to anoxia "…and redox conditions are consistently indicated by the PCA analysis (Fig. 5b)." has to be treated cautiously, as PCA analysis explains less than 50 % of the variance and the suggested interpretation might be a misinterpretation.

*We removed the geochemical data.*

**Comments to figures and tables**

Fig. 4: the positions of datapoints shown in 4b should be marked in 4a and the positions of datapoints shown in 4c in 4b. "Lithozone" is introduced here for a facies description. This is misleading in comparison to the term "units", which I would label as lithozones. Moreover, the lithology of Fig. 4a demonstrates that both cores are quite different. However, similarity is assumed throughout the manuscript. Here it would be nice to see how well the correlation really is by comparing MS data from both cores or using selected XRF data for this purpose.

We removed the zones and changed to "assemblages".

Fig. 8: c) the x-axis needs to be extended to ca. AD 1750, to show were the data meet with the x-axis. e) the Plum model needs to be explained. It cannot be understood as it is. d)+f) show calibrated 14-C ages in the topmost ca. 10 cm, data not listed in Tab. 3.

We extended the x-axis of (c). We added more text to explain the Plum model. We used different colors for panel d and f for 210Pb ages and 14C.

Tab. 1: minimum varve thickness is provided as 40-50 µm. This is just one silt grain and would be uncommonly thin for clastic varves. The unmentioned lake basin in the catchment could be an explanation for this phenomenon. However, this number is just a mere statistical value based on treated lamination measurements…

Yes, the minimum thickness is very small, but the sediments are predominantly clay-sized. The minimum value is derived from the modeled results but is informed by the thickness measurements. The method does not allow for a minimum value that is smaller than any measured lamination.

We had comment on the smaller basin in the catchment on line 92 "The lake is fed by a small pond and stream to the northwest and drained by Mill Creek to the northeast". This was not addressed further in the new manuscript.

Tab. 2: Please provide data (values and percentages) of how many "varves" were actually counted and measured and how many modelled.

We did not "count" varves. Please see responses made earlier to address this point.

Tab. 3: is the data of IonPlus 3528 a pMC age? If so, please mention this!

This was amended in the table.

Tab. A1: This table is difficult to understand. Perhaps the depth for each marker layer should be provided?

The table was amended to enhance clarity. To explain, for example, marker layer 1 for observer 1 was found at a "count" of 699 in COL17-2 and 660 in COL17-3, indicating a difference of 39 "counts".

Fig. A2: Please, add a scale.

A scale could not be added. We noted in the caption this figure is for illustrative purposes.

Fig. A3: This graphic is difficult to understand – there are three colors but only two explained in the legend.

The third color is the overlapped mixture of the other two. We made a note in the caption.

Fig. A4: What does it mean, if the "Ratio Mn/Fe is normalized to Ti counts"? Is this (Mn/Fe)/Ti? Usually, it does not make sense to normalize a ratio. Why it is done here needs to be explained under methods.

This figure was removed.

Fig. A6: For the correlation matrix information is lacking about which data (i.e. depth interval) is included in calculations. I assume, this is only the topmost 125 cm.

This figure was removed.

---

## Author Response (AR2)

**Responses to reviewers**

Thank you to the editor and reviewers 1 and 3 for your constructive comments. We have made changes described below to accommodate your suggestions. We continue to disagree with reviewer 2 about the definition of "varves" and, more importantly, about the utility and validity of our study. Zolitschka et al. (2015) define a varve as any annually laminated sediment. We have tried to further clarify that our study does not claim that every lamination in the Columbine Lake sediments are varves, rather we argue that the clear laminations are most likely annual, due to agreement of the counts of well-laminated sediment near the surface and the $^{137}$Cs date, and with the overall agreement with the $^{14}$C dates. Although chronological uncertainty remains, we strongly believe Columbine Lake sediments have varves, even though they are intermittently indistinct. We show in the manuscript that these layers are near annual through multiple independent dating methods. Alternative explanations that the laminations are near-annual (but of unknown origin) or that we've happened across what may be the first lake in this part of the world to have a pronounced Chernobyl $^{137}$Cs peak seems far less likely than our interpretation that Columbine Lake sediments have varves that are intermittently indistinct. The observation that these sediments do not meet the criteria for traditional varve counting is both clear in the text and the motivation for the manuscript. We have attempted to clarify in section 3.4, lines 147-156 try to address some of the issues brought up in the previous review regarding the definition of a varve. We have amended the text further to explain that the laminations in Columbine do not fit the definition of a varve couplet like those found in lakes with good quality sediment like one of the lakes mentioned, Skilak Lake. We reaffirm in the paragraph that one goal is to develop a method that can use sediment with indistinct laminations to build a chronology.

**Reviewer #1**

L82: missing subject at the end of the line?

The text was rephrased to say, "Laminated sediment, even when indistinct or intermittent, provides valuable information that can be used to improve chronologies and can provide new opportunities for regions that currently lack records (Ramisch et al., 2020)."

L158: first mention of R is earlier, so reference should be put in there – L125.

The reference was moved from line L158 to line L125.

L276: repetition?

Repeated word "contain" was removed.

L293: Reference after statement about true non-glacial clastic lamina.

A reference to Zolitschka et al (2015) was added to line 293.

L319: I do not follow the last sentence of this paragraph.

The sentence was trying to explain that the algorithm does not allow for values that are smaller than what was measured. The text was changed to make this clearer.

Table 1: While meaning is clear, the heading of column with variables is technically incorrect.

We have changed the heading to "summary statistic" and added a heading above COL17-2 and COL17-3 that says "core".

L442, Fig. 7 caption: No need for "Plum is a statistical…" it is already in the text.

Sentence was removed.

L453, Tab. 3 footnote: Is the last sentence missing something?

Yes, it was not clear. We have changed the sentence to say, "This date not used because it returned a modern age".

L467: Please consider changing the name, as reader must keep in mind that one integrated model means something different than the other depending on the position in text. Names should me more straightforward and unique.

Good point, we changed it to "multiple observers integrated model" or MOIM throughout the text. We also had to change the legend in Figure 8 so we provide an updated figure.

L515: Is "contain" a correct word choice?

We changed "contain" to "have".

L561-62: Last sentence should be rewritten.

We changed the last sentence to "We consider this a significant advantage of this approach, as it objectifies the subjective element of observer judgement, puts less emphasis on the observers, and tends to align discrepancies."

L584: Safer to use "often" than "generally".

Changed "generally" to "often".

L599: With this sentence in the text one can expect short explanation and provided reasons why such analysis was not undertaken.

We deleted this as it was not relevant.

Tab. A2: same comment as for Tab. 3. Either way, consistent naming of headers.

We have changed the header to be consistent with Tab 1. We added a row above the core names and changed the header to "statistics".

Fig. A2: Can you provide scale bars?

Thank you for requesting those. We had to remake the figure from scratch to get the scale bars to show up which is why the images are not the same as in the previous manuscript version.

**Reviewer #3**

Line 168 - "three" needs to be changed to "four", according to the list in the previous lines

Good catch, "three" was changed to "four".

Line 179 - Either remove the comma after "probability" or remove both parentheses

Comma was removed.

Line 187 to 189 - I would suggest either removing the numbers in the brackets or saying directly that the laminations are assigned one of six different codes. As I understand it, in your current version of the manuscript you want to introduce the idea of prior probability estimates for over- and under-counting, but for example laminations with code 4 are also assigned a probability for over-counting (reference to line 192).

Yes, exactly. Thank you for providing clearer language that we can use. We have changed the text to now say on line 187-189 "Each lamination was assigned one of six different codes (Appendix A Fig. A2) with a corresponding distribution of over and under-counting prior probability estimate (Sect. 3.5). Codes 1, 2 and 3 are assigned by the clarity of the lamination's appearance, with a code value of 1 being of higher clarity than a code value of 3."

Line 210 to 213 - I understand the reason for assigning code 5 and 6 to specific indistinct core sections and then using an emulator. But how can the user influence the modeling process if the indistinct sections are due to a change in the sedimentation process? Would the emulator still try to match the missing section with a surrogate section?

Yes, based on the assumption that sediment processes are stable. If there are changes in the sediment process during those intervals, the emulator would be wrong. We changed the text to make this clearer. Lines 208-211 now say "This approach assumes that the sedimentation processes in these intervals is consistent with the well laminated sections and other laminated intervals can serve as surrogates for indistinct sections. We argue that this assumption is valid for Columbine Lake, as the distribution of the lamination thickness is similar in both cores throughout the sections with distinct laminations (Appendix A Fig. A3)."

Subsection 3.5 (line 215 to 225) - What happens to the laminations labeled 4 to 6 in the varve-only model? I think this needs further clarification on how these laminations are then incorporated in the modeling process.

Yes, good point. Basically codes 4-6 were treated the same in both scenarios. Only codes 1-3 changed. We changed the text on lines 218-221 to now say "In both scenarios, codes 1, 2, and 3 were given over- and under-counting priors, code 4 was given a 50% chance of over-counting and a 0% chance of under-counting, and codes 5 and 6 were simulated using the emulator as described above. Codes 4-6 were treated the same in both scenarios, and only codes 1-3 changed. In the first scenario, the priors for codes 1-3 were symmetrical and based on values found in the literature (Fig. 2a, e.g., Dräger et al., 2017)."

Line 256 - Since you have already introduced Plum, I would shorten the sentence to "[…] that use conventional CRS, CFCS, or Plum (Sec.3.2)."

We changed the text as suggested.

Line 427 - At the end of the line you wrote "model" twice.

One word was removed.

**Reviewer #2**

This review is about the revised version of the manuscript by Arcusa et al. investigating non-glacial laminated clastic sediments from high-mountain (3874 m asl) Columbine Lake in Colorado (USA). The study focusses on lamination counting in combination with statistical data treatment for the establishment of an age/depth model supported by radiometric dating. The obtained record of sedimentation rates is intended to be used for future environmental studies of this lacustrine archive.

The revision benefits from geochemical and paleoenvironmental data and interpretations being excluded. However, the conclusions drawn are still premature and highly disputable. Unfortunately, the authors did not consider major parts of my previous review, thus they are insufficiently taking care of criticism related to the chronology. I am not repeating these still valid arguments, but add some comments and suggestions that might help the authors to progress with this study.

In general, the dating of clastic sediment sequences is difficult. Not only because organic material for radiocarbon dating is rare or lacking, but also because depositional processes are manifold and especially in high mountain regions dependent on complex and highly variable lake internal (redeposition, sediment focussing) and catchmentrelated sediment transfer mechanisms, which are closely linked to snowmelt, melting lake ice, precipitation and (currently not in this case) to glacier melt. Not considered by the authors is the existence of a delta, which indicates that quite some amount of sediment reached the lake, which disagrees with rather low sedimentation rates reconstructed. Moreover, depositional processes are not well understood. For instance, anoxia are suspected for the deep basin (l. 92) as necessary for varve preservation but not supported by any data. However, such conditions are very unrealistic for an oligotrophic high mountain lake with clastic sediments.

Moreover, there is still no proof of concept documenting that the observed layers are annually laminated. The authors argue that the Cs peak agrees with the "varve chronology". But this is weak evidence if at all, as the annual character of laminations is a mere hypothesis, indistinct laminations were interpolated, some cases of discontinuity exist as well as erosion. Thus, Pb dates are a more realistic approach to verify either Chernobyl or 1963 for the Cs peak. Following this approach, Chernobyl is the choice with an age range between CE 1984-1996 based on Pb dating. The argument that no other records with a Chernobyl signal have been reported for North America so far is not acceptable. This may be the first record and the reason may be related to the high elevation of this site?!

The authors argue that their "method does not ´count´ varves in the traditional sense of varve `counting`." But there is no further explanation what is the way they have chosen! As I understand figures and tables, they do count the laminations as done by others before. In line 150-51 it is stated that the term "varve" relates in this study to an algorithm-modelled or algorithm-simulated annually deposited lamination and not to the sediment record itself – thus being a theoretical description for the model output. It remains unclear, however, how the images of laminations are fed into the model and how the algorithm is then able to combine two or more of them to become a varve, i.e. integrating time into the record. Furthermore, this (scientifically inacceptable) discrimination between laminations and varves is not consistently applied. One out of many examples is the title of the paper: "...varve measurements in intermittently varved sediments". In both cases, "varve" is here related to the sediment and not to the model output! In line 212-13 it is suggested that "...the indistinct laminations are due to changes in preservation, not the sedimentation process." – an unsupported hypothesis. Also, the merging of the radiocarbon chronology with the lamination-based age/depth model to form the final output – the integrated chronology – is critical methodologically but also due to a problematic radiocarbon chronology. Explanation: Clastic sediments hardly contain organic matter. Thus, for radiocarbon dating the MICADAS is applied, which was developed to date very small sample sizes. But this easily causes errors, especially if the samples are extremely small (authors did not provide the weight of analysed material) and if samples are a mixture of different organic fractions. Moreover, contamination is a problem as well as reservoir effects (reworked organics from the catchment area and/or the littoral zone are frequent for clastic sediment records as in this case – see Tab. 3) may be present in addition leading to radiocarbon dates being too old and explaining the low sedimentation rates for a clastic depositional system with inflow. Finally, 78 % of the record has a poor varve quality code (according to their nomenclature: this should be a lamination quality!).

In conclusion, if the authors provide no evidence that counted laminations are annual, how can the algorithm reproduce an annually laminated depositional system (l. 253-54)? The "integrated radiometric-varve chronology" (l. 614) is a modification (modelling/simulation) of lamination counts to best fit the radiometric age/depth model. For such an age/depth model the term varve chronology is not applicable. At best it might be called a sedimentation rate-corrected radiocarbon chronology. The quoted sites of Lake Suigetsu and Skilak in the introduction are both dominantly varved sediment records and their annual character was determined convincingly. Lake Columbine with its large amount of non- or intermittently laminated sediment and without any characterisation of laminations being annual does not seem to be a good example to carry out the described modelling approach.